# LEARNING LINEAR STATE-SPACE MODELS WITH SPARSE SYSTEM MATRICES

**Yasen Wang**[1,2][*]**, Kaiqi Fang**[3]**, Guijun Ma** [3,4][*]**,**
**Junlin Li**[5]**, Mengyu Sun**[6]**, Zhilan Huang**[1]**, Gang Lu**[1]

[1]China Telecom Research Institute, Guangzhou, China
[2]School of Future Technology, South China University of Technology, Guangzhou, China
[3]School of Artificial Intelligence and Automation,
 Huazhong University of Science and Technology, Wuhan, China
[4]State Key Laboratory of Intelligent Manufacturing Equipment and Technology,
 Huazhong University of Science and Technology, Wuhan, China
[5]School of Mathematics and Statistics, Fuyang Normal University, Fuyang, China
[6]China Telecom Research Institute, Beijing, China

## ABSTRACT

Due to tractable analysis and control, linear state-space models (LSSMs) provide a fundamental mathematical tool for time-series data modeling in various disciplines. In particular, many LSSMs have sparse system matrices because interactions among variables are limited or only a few significant relationships exist. However, current learning algorithms for LSSMs lack the ability to learn system matrices with the sparsity constraint due to the similarity transformation. To address this issue, we impose sparsity-promoting priors on system matrices to balance modeling error and model complexity. By taking hidden states of LSSMs as latent variables, we then explore the expectation–maximization (EM) algorithm to derive a maximum a posteriori (MAP) estimate of both hidden states and system matrices from noisy observations. Based on the Global Convergence Theorem, we further demonstrate that the proposed learning algorithm yields a sequence converging to a local maximum or saddle point of the joint posterior distribution. Finally, experimental results on simulation and real-world problems illustrate that the proposed algorithm can preserve the inherent topological structure among variables and significantly improve prediction accuracy over classical learning algorithms.

## 1 INTRODUCTION

Linear state-space models (LSSMs) are fundamental mathematical tools for analyzing time-series data with applications in robotics (Mamakoukas et al., 2019; 2020), systems biology (Jin et al., 2020b; Pillonetto & Ljung, 2023), and natural language processing (Smith et al., 1999; Belanger & Kakade, 2015). Generally, LSSMs describe time-series data $\{(\boldsymbol{u}_t, \boldsymbol{y}_t)\}_{t=1}^{T}$ through the following stochastic difference equation:

$$\boldsymbol{x}_t = \boldsymbol{A}\boldsymbol{x}_{t-1} + \boldsymbol{B}\boldsymbol{u}_t + \boldsymbol{\varepsilon}_t, \tag{1}$$

$$\boldsymbol{y}_t = \boldsymbol{C}\boldsymbol{x}_t + \boldsymbol{D}\boldsymbol{u}_t + \boldsymbol{\omega}_t, \tag{2}$$

where $\boldsymbol{u}_t \in \mathbb{R}^p$ is the input signal, $\boldsymbol{y}_t \in \mathbb{R}^m$ is the noisy observation, $\boldsymbol{x}_t \in \mathbb{R}^n$ is the hidden state, $\boldsymbol{A} \in \mathbb{R}^{n \times n}$, $\boldsymbol{B} \in \mathbb{R}^{n \times p}$, $\boldsymbol{C} \in \mathbb{R}^{m \times n}$, and $\boldsymbol{D} \in \mathbb{R}^{m \times p}$ are the unknown system matrices, and $\boldsymbol{\varepsilon}_t \sim \mathcal{N}(\boldsymbol{0}, \boldsymbol{R})$ and $\boldsymbol{w}_t \sim \mathcal{N}(\boldsymbol{0}, \boldsymbol{Q})$ are the diagonal process and measurement noise, respectively. In addition, LSSMs are also widely used to approximate complex non-linear systems in industrial processes given their relative simplicity (Yuan et al., 2017; Lusch et al., 2018). Due to a complete rigorous theory available on LSSMs, learning them from noisy observations can enable us to make tractable analysis and control of systems (Chen & Poor, 2022; Bakshi et al., 2023).

---

[*]Correspondence to: Yasen Wang (arthinw@hust.edu.cn), Guijun Ma (mgj@hust.edu.cn). Source codes are available on GitHub at `https://github.com/ArthinYS/Learning-Sparse-LSSM`.

In this paper, we focus on learning LSSMs with sparse system matrices for two important reasons. First, the learned LSSMs should include the minimally required parameters to explain time-series data following the *Occam's razor* principle, which favors explanations constructed with the smallest possible set of elements. Additionally, many real-world systems indeed have a sparse topology, as each state or measurement variable only depends on a few other state variables and inputs (Efroni et al., 2022). For example, a gene only regulates the expression of a limited number of other genes in gene regulatory networks (He et al., 2024b). In industry, communication systems usually have a sparse topology to reduce energy consumption (Jin et al., 2020a;b). However, available learning algorithms lack the ability to learn LSSMs with the sparsity constraint on system matrices due to the similarity transformation.

To learn LSSMs with sparse system matrices, we impose sparsity-promoting priors on them to balance model complexity and modeling error. Following the Bayes' rule, we can combine the marginal likelihood and prior functions to derive the joint posterior distribution of all unknown variables. However, directly maximizing such a posterior distribution to estimate system matrices is intractable because the hidden states of LSSMs are unknown. To address this issue, we explore the expectation–maximization (EM) algorithm to give an alternate maximum a posteriori (MAP) estimate of hidden states and system matrices by taking hidden states as latent variables. In the expectation step, we use the Rauch–Tung–Striebel (RTS) smoother to give a closed-form update rule for the hidden states. In the maximization step, we leverage the block coordinate descent method to analytically update the system matrices in turn. By alternately performing the expectation and maximization steps until convergence, the proposed algorithm can determine the sparse system matrices of LSSMs from noisy observations. In summary, the contributions of this paper are threefold:

- Leveraging sparsity-promoting techniques, we propose an algorithm to learn LSSMs with sparse system matrices from noisy observations. Following the Global Convergence Theorem (Luenberger et al., 1984), we also demonstrate that the proposed algorithm is guaranteed to converge to a local maximum or saddle point of the posterior distribution composed of the marginal likelihood and prior functions.

- Because available learning algorithms only learn LSSMs up to a similarity transformation, the learned system matrices usually differ from the true ones in both numerical values and topological structure. However, the proposed algorithm learns system matrices by balancing model complexity and modeling error. As a result, the learned system matrices can preserve the inherent topological structure among variables, which is a significant improvement over classical learning algorithms.

- Experimental results on simulation and real-world datasets demonstrate that the proposed algorithm outperforms classical ones on learning LSSMs with sparse system matrices. In addition, the learned system matrices of the proposed algorithm are more valuable for exploring the interaction laws of systems.

## 2 RELATED WORK

**Least-squares minimization.** Basically, least-squares minimization (LSM) learns unknown models by minimizing the sum of the squares of residuals (Faradonbeh et al., 2018; 2020; Modi et al., 2024). By taking the one-step prediction error as the objective function in LSM, prediction error minimization (PEM) is proposed to learn LSSMs via gradient-based optimization methods (Ljung, 2002; Katayama et al., 2005). Given a symmetric transition matrix, Hazan et al. (2017) design an efficient method for the online prediction of LSSMs by formulating system identification as an online PEM problem. Recently, combining the Ho–Klamn (HK) algorithm with LSM, Oymak & Ozay (2019) propose a method to learn system matrices of LSSMs with sample complexity analysis. However, it is well known that LSM is sensitive to noise and cannot characterize the sparsity of system matrices (Tibshirani, 1996; Martens, 2010).

**Subspace state-space system identification.** Subspace state-space system identification (4SID) algorithms project data Hankel matrices onto certain subspaces to estimate the extended observability matrix and hidden states using linear algebra tools (Larimore, 1990; Verahegen & Dewilde, 1992; Van Overschee & De Moor, 1994; He et al., 2024a). Subsequently, system matrices can be recovered from either the extended observability matrix or hidden states (Favoreel et al., 2000). Based on principal component analysis, Wang & Qin (2002) present a new 4SID algorithm to learn LSSMs under

the errors-in-variables situation. By choosing different weighting matrices to perform the singular value decomposition, Van Overschee & De Moor (2012) provide a geometric framework to unify almost all classical 4SID methods. Further, Huang et al. (2016) present the Weight-Least-Square method to learn stable LSSMs by multiplying the unstable component with a weight matrix. However, it is widely recognized that such algorithms generally cannot obtain accurate system matrices as required (Martens, 2010; Qin, 2006).

**Maximum likelihood estimation.** Because the joint likelihood function of LSSMs involves hidden states, the EM algorithm is employed to give the maximum likelihood estimation (MLE) of system matrices (Shumway & Stoffer, 1982; Ghahramani & Hinton, 1996). Leveraging the EM algorithm, the distribution of hidden states can be explicitly derived using the Kalman smoother based on the current estimate of system matrices. It then updates system matrices by maximizing the expected log-likelihood with respect to the hidden states. To present a robust MLE for LSSMs, Gibson & Ninness (2005) implement the expectation and maximization steps via the LR and Cholesky factorisation, respectively. To increase the efficiency of EM for learning LSSMs, Martens (2010) proposes an approximate second-order statistics (ASOS) scheme to approximate the expectation step. Combining EM and Lagrangian relaxation, Umenberger et al. (2018) use semidefinite programming to optimize the tight bounds on the likelihood to learn LSSMs with model stability constraints. However, such learning algorithms lack the ability to deal with sparse system matrices.

**Sparsity-promoting methods.** By adding a penalty term on model parameters, sparsity-promoting methods can balance model complexity and modeling error to learn systems from data (Brunton et al., 2016). Leveraging the $\ell_1$ regularization term, Tibshirani (1996) proposes a method named Lasso to estimate parameters in linear models. Further, reweighted $\ell_1$ minimization is proposed to enhance sparsity (Wipf & Nagarajan, 2007; Candes et al., 2008; Garrigues & Olshausen, 2010). However, solving an $\ell_1$ minimization problem is challenging due to its non-differentiability at the origin, and these methods also require careful fine-tuning of hyperparameters. To address such issues, sparse Bayesian learning (SBL) imposes sparsity-promoting priors on model parameters to enforce sparsity (Samanta et al., 2022; Chakraborty et al., 2023). Subsequently, it maximizes the posterior distribution consisting of the likelihood function and priors to estimate model parameters and hyperparameters (Tipping, 2001; Wipf & Rao, 2004). Recently, SBL has been applied to learn various systems from data, with system states being measurable yet potentially corrupted by process noise (Pan et al., 2015; Yuan et al., 2019; Wang et al., 2024). However, leveraging such sparsity-promoting methods to learn LSSMs with sparse system matrices remains an elusive and challenging problem because system states are unavailable and observed data are corrupted by both process and measurement noise (Course & Nair, 2023).

## 3 METHODOLOGY

Due to the similarity transformation, LSSMs admit many equivalent representations with different levels of sparsity, where the corresponding transformed system matrices are given by $\boldsymbol{\Phi} \boldsymbol{A} \boldsymbol{\Phi}^{-1}$, $\boldsymbol{\Phi} \boldsymbol{B}$, $\boldsymbol{C} \boldsymbol{\Phi}^{-1}$, and $\boldsymbol{D}$, with $\boldsymbol{\Phi} \in \mathbb{R}^{n \times n}$ being a nonsingular matrix. However, we focus on learning the LSSMs with sparse system matrices that include minimally required parameters in accordance with the *Occam's razor* principle. Hence, we define the identifiability of LSSMs with sparse system matrices to ensure that the resulting ambiguities can only be permutations and scaling, as formalized below.

**Definition 3.1.** *(Identifiability) For LSSMs with nonzero system matrices $\boldsymbol{A}, \boldsymbol{B}, \boldsymbol{C},$ and $\boldsymbol{D}$, if any nonsingular matrix $\boldsymbol{\Phi} \in \mathbb{R}^{n \times n}$ satisfying*

$$\|\boldsymbol{\Phi} \boldsymbol{A} \boldsymbol{\Phi}^{-1}\|_0 = \|\boldsymbol{A}\|_0, \quad \|\boldsymbol{\Phi} \boldsymbol{B}\|_0 = \|\boldsymbol{B}\|_0, \quad \text{and} \quad \|\boldsymbol{C} \boldsymbol{\Phi}^{-1}\|_0 = \|\boldsymbol{C}\|_0, \tag{3}$$

*must be a generalized permutation matrix, then such systems are said to be essentially identifiable, up to permutation and scaling.*

### 3.1 STUDENT'S $t$-DISTRIBUTION PRIOR

Here, we impose the Student's $t$-distribution prior on the system matrices $\boldsymbol{A}, \boldsymbol{B}, \boldsymbol{C},$ and $\boldsymbol{D}$ to promote model sparsity, because it can be sharply peaked at zero compared to other priors (Tipping, 2001). Generally, the Student's $t$-distribution prior is implemented in a hierarchical way (Wang et al., 2024; Zhou et al., 2021). It imposes a zero-mean Gaussian prior on the system matrices and

then adopts an Inverse-Gamma distribution on the unknown variance. For example, we can impose the Student's $t$-distribution prior on $\boldsymbol{A}$ to promote its sparsity as follows:

$$p(\boldsymbol{A} \mid \boldsymbol{\Gamma}_a) = \prod_{i=1}^{n} \prod_{j=1}^{n} p\left(\boldsymbol{A}_{ij} \mid \boldsymbol{\Gamma}_{a,ij}\right) = \prod_{i=1}^{n} \prod_{j=1}^{n} \frac{1}{\sqrt{2\pi \boldsymbol{\Gamma}_{a,ij}}} \exp\left(-\frac{\boldsymbol{A}_{ij}^2}{2\boldsymbol{\Gamma}_{a,ij}}\right), \tag{4}$$

$$p(\boldsymbol{\Gamma}_a) = \prod_{i=1}^{n} \prod_{j=1}^{n} \frac{a_0^{b_0}}{\Gamma(a_0)} \boldsymbol{\Gamma}_{a,ij}^{-a_0-1} \exp\left(-\frac{b_0}{\boldsymbol{\Gamma}_{a,ij}}\right), \tag{5}$$

where $\Gamma(\cdot)$ is the gamma function, and $\boldsymbol{A}_{ij}$ and $\boldsymbol{\Gamma}_{a,ij}$ are the $ij$th components of $\boldsymbol{A}$ and $\boldsymbol{\Gamma}_a$, respectively. To generate non-informative hyperprior on $\boldsymbol{\Gamma}_{a,ij}$, $a_0$ and $b_0$ are typically set to very small values (e.g., $10^{-6}$). In addition, $\boldsymbol{\Gamma}_b$, $\boldsymbol{\Gamma}_c$, $\boldsymbol{\Gamma}_d$, $\boldsymbol{\Gamma}_{b,ij}$, $\boldsymbol{\Gamma}_{c,ij}$, and $\boldsymbol{\Gamma}_{d,ij}$ are defined in a similar manner (see Appendix A). For the process noise $\boldsymbol{R}$ and measurement noise $\boldsymbol{Q}$, we impose a uniform distribution prior on them to derive a flat prior.

## 3.2 LOSS FUNCTION

Following the Bayes' rule, we can combine the marginal likelihood and prior functions to derive the joint posterior distribution of all the unknown variables as follows:

$$p(\boldsymbol{\Theta} \mid \boldsymbol{Y}) \propto \underbrace{p(\boldsymbol{Y} \mid \boldsymbol{\Theta})}_{\text{marginal likelihood}} \times \underbrace{p(\boldsymbol{\Theta})}_{\text{prior}}, \tag{6}$$

where $\boldsymbol{\Theta} = \{\boldsymbol{A}, \boldsymbol{B}, \boldsymbol{C}, \boldsymbol{D}, \boldsymbol{R}, \boldsymbol{Q}, \boldsymbol{\Gamma}_a, \boldsymbol{\Gamma}_b, \boldsymbol{\Gamma}_c, \boldsymbol{\Gamma}_d\}$ is the set of unknown variables and $\boldsymbol{Y} = [\boldsymbol{y}_1, \boldsymbol{y}_2, ..., \boldsymbol{y}_T]$. Because the system state $\boldsymbol{x}_t$ is unobserved, it is hard to explicitly compute $p(\boldsymbol{Y} \mid \boldsymbol{\Theta})$. Hence, directly maximizing equation 6 to estimate $\boldsymbol{\Theta}$ is generally intractable. To tackle this problem, we explore the EM algorithm to iteratively improve equation 6 by regarding $\boldsymbol{x}_t$ as the latent variable. Instead of directly maximizing equation 6, the EM algorithm focuses on improving the expected value of the log posterior function of $\boldsymbol{\Theta}$ with respect to the state vector $\boldsymbol{X} = [\boldsymbol{x}_1, \boldsymbol{x}_2, ..., \boldsymbol{x}_T]$ as follows:

$$H(\boldsymbol{\Theta} \mid \boldsymbol{\Theta}^k) = \mathbb{E}_{\boldsymbol{X}^k}[\log\left(p(\boldsymbol{Y}, \boldsymbol{X} \mid \boldsymbol{\Theta})p(\boldsymbol{\Theta})\right)], \tag{7}$$

where $\boldsymbol{X}^k \sim p\left(\boldsymbol{X} \mid \boldsymbol{Y}, \boldsymbol{\Theta}^k\right)$, and $\boldsymbol{X}^k$ and $\boldsymbol{\Theta}^k$ denote the estimates of $\boldsymbol{X}$ and $\boldsymbol{\Theta}$ at the $k$th iteration, respectively. It is well-known that iteratively maximizing equation 7 is equivalent to iteratively maximizing equation 6 (Little & Rubin, 2019).

## 3.3 EXPECTATION STEP: RAUCH–TUNG–STRIEBEL SMOOTHER

Because equation 7 involves $p\left(\boldsymbol{X} \mid \boldsymbol{Y}, \boldsymbol{\Theta}^k\right)$, we first need to derive the conditional distribution of $\boldsymbol{x}_t$ given $\boldsymbol{Y}$ and current $\boldsymbol{\Theta}^k = \{\boldsymbol{A}^k, \boldsymbol{B}^k, \boldsymbol{C}^k, \boldsymbol{D}^k, \boldsymbol{R}^k, \boldsymbol{Q}^k, \boldsymbol{\Gamma}_a^k, \boldsymbol{\Gamma}_b^k, \boldsymbol{\Gamma}_c^k, \boldsymbol{\Gamma}_d^k\}$, which can be formulated as a classical smoothing problem. For LSSMs, the RTS smoother provides a closed-form smoothing solution for $p\left(\boldsymbol{x}_t \mid \boldsymbol{Y}, \boldsymbol{\Theta}^k\right)$.

**Lemma 3.1.** *(RTS smoother Särkkä & Svensson (2023)) For LSSMs, the RTS smoother states that*

$$p\left(\boldsymbol{x}_t \mid \boldsymbol{Y}, \boldsymbol{\Theta}^k\right) = \mathcal{N}\left(\boldsymbol{m}_t^k, \boldsymbol{P}_t^k\right), \tag{8}$$

*where $t = 0, ..., T$. Here, $\boldsymbol{m}_t^k$ and $\boldsymbol{P}_t^k$ are derived via the reverse-time recursions as follows:*

$$\boldsymbol{m}_t^k = \boldsymbol{\mu}_t^k + \boldsymbol{G}_t^k\left(\boldsymbol{m}_{t+1}^k - \overline{\boldsymbol{\mu}}_{t+1}^k\right), \tag{9}$$

$$\boldsymbol{P}_t^k = \boldsymbol{\Sigma}_t^k + \boldsymbol{G}_t^k\left(\boldsymbol{P}_{t+1}^k - \overline{\boldsymbol{\Sigma}}_{t+1}^k\right)\left(\boldsymbol{G}_t^k\right)', \tag{10}$$

*where $\boldsymbol{G}_t^k = \boldsymbol{\Sigma}_t^k\left(\boldsymbol{A}^k\right)'\left(\overline{\boldsymbol{\Sigma}}_{t+1}^k\right)^{-1}$. The quantities $\boldsymbol{\mu}_t^k$, $\overline{\boldsymbol{\mu}}_t^k$, $\boldsymbol{\Sigma}_t^k$, and $\overline{\boldsymbol{\Sigma}}_t^k$ coupled in equation 9 and equation 10 are pre-computed using the Kalman filter as follows:*

$$\overline{\boldsymbol{\mu}}_t^k = \boldsymbol{A}^k \boldsymbol{\mu}_{t-1}^k + \boldsymbol{B}^k \boldsymbol{u}_t, \quad \overline{\boldsymbol{\Sigma}}_t^k = \boldsymbol{A}^k \boldsymbol{\Sigma}_{t-1}^k\left(\boldsymbol{A}^k\right)' + \boldsymbol{R}^k, \tag{11}$$

$$\boldsymbol{K}_t^k = \overline{\boldsymbol{\Sigma}}_t^k\left(\boldsymbol{C}^k\right)'\left(\boldsymbol{C}^k \overline{\boldsymbol{\Sigma}}_t^k\left(\boldsymbol{C}^k\right)' + \boldsymbol{Q}^k\right)^{-1}, \tag{12}$$

$$\boldsymbol{\mu}_t^k = \overline{\boldsymbol{\mu}}_t^k + \boldsymbol{K}_t^k\left(\boldsymbol{Y}_t - \boldsymbol{C}^k \overline{\boldsymbol{\mu}}_t^k - \boldsymbol{D}^k \boldsymbol{u}_t\right), \quad \boldsymbol{\Sigma}_t^k = \left(\boldsymbol{I}_n - \boldsymbol{K}_t^k \boldsymbol{C}^k\right)\overline{\boldsymbol{\Sigma}}_t^k, \tag{13}$$

*where $\boldsymbol{I}_n$ is an identity matrix of dimension $n$. Note that the reverse-time recursions of equation 9 and equation 10 start from the initial conditions $\boldsymbol{m}_T^k = \boldsymbol{\mu}_T^k$ and $\boldsymbol{P}_T^k = \boldsymbol{\Sigma}_T^k$, and the recursions of equation 11–equation 13 start from the mean $\boldsymbol{\mu}_0^k$ and covariance $\boldsymbol{\Sigma}_0^k$ of the initial state $\boldsymbol{x}_0$.*

Besides $p\left(\boldsymbol{x}_t \mid \boldsymbol{Y}, \boldsymbol{\Theta}^k\right)$, we also need to derive the covariance matrix between the adjacent states $\boldsymbol{x}_t$ and $\boldsymbol{x}_{t-1}$ given $\boldsymbol{Y}$ and $\boldsymbol{\Theta}^k$ to compute equation 7. To address this issue, the following lemma gives necessary recursions.

**Lemma 3.2.** *(The lag-one covariance smoother Särkkä & Svensson (2023)) For LSSMs, the covariance matrix $\boldsymbol{P}_{t,t-1}^k$ between the adjacent states $\boldsymbol{x}_t$ and $\boldsymbol{x}_{t-1}$ given $\boldsymbol{Y}$ and $\boldsymbol{\Theta}^k$ can be recursively derived as follows:*

$$\boldsymbol{P}_{t,t-1}^k = \left(\boldsymbol{\Sigma}_t^k + \boldsymbol{G}_t^k \boldsymbol{P}_{t+1,t}^k - \boldsymbol{G}_t^k \boldsymbol{A}^k \boldsymbol{\Sigma}_t^k\right)\left(\boldsymbol{G}_{t-1}^k\right)' \tag{14}$$

*with $\boldsymbol{P}_{T,T-1}^k = \left(\boldsymbol{I}_n - \boldsymbol{K}_T^k \boldsymbol{C}^k\right)\boldsymbol{A}^k \boldsymbol{\Sigma}_{T-1}^k$.*

Based on Lemmas 3.1 and 3.2, we are able to calculate the loss function in equation 7 as follows:

$$H\left(\boldsymbol{\Theta} \mid \boldsymbol{\Theta}^k\right) = H_1\left(\boldsymbol{A}, \boldsymbol{B}, \boldsymbol{R}\right) + H_2(\boldsymbol{C}, \boldsymbol{D}, \boldsymbol{Q}) + H_3(\boldsymbol{A}, \boldsymbol{B}, \boldsymbol{C}, \boldsymbol{D}, \boldsymbol{\Gamma}_a, \boldsymbol{\Gamma}_b, \boldsymbol{\Gamma}_c, \boldsymbol{\Gamma}_d), \tag{15}$$

where

$$H_1(\boldsymbol{A}, \boldsymbol{B}, \boldsymbol{R}) = \mathbb{E}_{\boldsymbol{X}^k}[\log p(\boldsymbol{X}|\boldsymbol{A}, \boldsymbol{B}, \boldsymbol{R})], \tag{16}$$

$$H_2(\boldsymbol{C}, \boldsymbol{D}, \boldsymbol{Q}) = \mathbb{E}_{\boldsymbol{X}^k}[\log p(\boldsymbol{Y}|\boldsymbol{X}, \boldsymbol{C}, \boldsymbol{D}, \boldsymbol{Q})], \tag{17}$$

$$H_3(\boldsymbol{A}, \boldsymbol{B}, \boldsymbol{C}, \boldsymbol{D}, \boldsymbol{\Gamma}_a, \boldsymbol{\Gamma}_b, \boldsymbol{\Gamma}_c, \boldsymbol{\Gamma}_d) = \mathbb{E}_{\boldsymbol{X}^k}[\log\left(p(\boldsymbol{A}, \boldsymbol{\Gamma}_a)p(\boldsymbol{B}, \boldsymbol{\Gamma}_b)p(\boldsymbol{C}, \boldsymbol{\Gamma}_c)p(\boldsymbol{D}, \boldsymbol{\Gamma}_d)\right)]. \tag{18}$$

Due to the limited space, the detailed derivation of equation 15 and explicit mathematical expressions of $H_1(\boldsymbol{A}, \boldsymbol{B}, \boldsymbol{R})$, $H_2(\boldsymbol{C}, \boldsymbol{D}, \boldsymbol{Q})$, and $H_3(\boldsymbol{A}, \boldsymbol{B}, \boldsymbol{C}, \boldsymbol{D}, \boldsymbol{\Gamma}_a, \boldsymbol{\Gamma}_b, \boldsymbol{\Gamma}_c, \boldsymbol{\Gamma}_d)$ are given in Appendix B.1.

## 3.4 MAXIMIZATION STEP: BLOCK COORDINATE DESCENT

Obviously, $H\left(\boldsymbol{\Theta} \mid \boldsymbol{\Theta}^k\right)$ is a non-convex function and unknown variables are highly coupled. To provide analytical update formulas, we leverage the block coordinate descent method to sequentially optimize the model parameters.

**Update procedures of $\boldsymbol{A}$, $\boldsymbol{B}$, $\boldsymbol{C}$, and $\boldsymbol{D}$.** Due to the introduction of sparsity-promoting priors, it is intractable to derive closed-form solutions for $\boldsymbol{A}$, $\boldsymbol{B}$, $\boldsymbol{C}$, and $\boldsymbol{D}$ by setting the derivatives of $H\left(\boldsymbol{\Theta} \mid \boldsymbol{\Theta}^k\right)$ with respect to these variables to zero directly. To address this issue, we adopt a row-wise update rule for $\boldsymbol{A}$, $\boldsymbol{B}$, $\boldsymbol{C}$, and $\boldsymbol{D}$. For example, the derivative of $H\left(\boldsymbol{\Theta} \mid \boldsymbol{\Theta}^k\right)$ with respect to the $r$th row of $\boldsymbol{A}$, denoted as $\boldsymbol{A}_r$, is as follows:

$$\frac{\partial\left(H_1(\boldsymbol{A}, \boldsymbol{B}^k, \boldsymbol{R}^k) + H_3(\boldsymbol{A}, \boldsymbol{B}^k, \boldsymbol{C}^k, \boldsymbol{D}^k, \boldsymbol{\Gamma}_a^k, \boldsymbol{\Gamma}_b^k, \boldsymbol{\Gamma}_c^k, \boldsymbol{\Gamma}_d^k)\right)}{\partial \boldsymbol{A}_r}$$

$$= \sum_{t=1}^T \left(\boldsymbol{R}_{rr}^k\right)^{-1}\left(\left(\boldsymbol{m}_{t,r}^k - \boldsymbol{A}_r \boldsymbol{m}_{t-1}^k - \boldsymbol{B}_r^k \boldsymbol{u}_t\right)\left(\boldsymbol{m}_{t-1}^k\right)' + \left(\boldsymbol{P}_{t,t-1,r}^k - \boldsymbol{A}_r \boldsymbol{P}_{t-1}^k\right)\right) - \boldsymbol{A}_r \overline{\boldsymbol{\Gamma}}_{a,r}^{kd}, \tag{19}$$

where $\boldsymbol{R}_{rr}^k$ is the $rr$th component of $\boldsymbol{R}^k$, $\boldsymbol{P}_{t,t-1,r}^k$, $\boldsymbol{m}_{t,r}^k$, and $\boldsymbol{B}_r^k$ are the $r$th rows of $\boldsymbol{P}_{t,t-1}^k$, $\boldsymbol{m}_t^k$, and $\boldsymbol{B}^k$, respectively. In particular, $\overline{\boldsymbol{\Gamma}}_{a,r}^{kd} = \text{diag}[\overline{\boldsymbol{\Gamma}}_{a,r}^k]$ with $\overline{\boldsymbol{\Gamma}}_{a,r}^k$ being the $r$th row of $\overline{\boldsymbol{\Gamma}}_a^k$. Setting equation 19 to zero leads to the following update rule for $\boldsymbol{A}_r$ at the $k$th iteration:

$$\boldsymbol{A}_r^{k+1} = \left(\sum_{t=1}^T \left(\left(\boldsymbol{m}_{t,r}^k - \boldsymbol{B}_r^k \boldsymbol{u}_t\right)\left(\boldsymbol{m}_{t-1}^k\right)' + \boldsymbol{P}_{t,t-1,r}^k\right)\right)$$

$$\times \left(\sum_{t=1}^T \left(\boldsymbol{P}_{t-1}^k + \boldsymbol{m}_{t-1}^k\left(\boldsymbol{m}_{t-1}^k\right)'\right) + \boldsymbol{R}_{rr}^k \overline{\boldsymbol{\Gamma}}_{a,r}^{kd}\right)^{-1}. \tag{20}$$

The detailed derivation of equation 19 can be found in Appendix B.2. Similarly, we can update the $r$th row of $\boldsymbol{B}$, $\boldsymbol{C}$, and $\boldsymbol{D}$ at the $k$th iteration as follows:

$$\boldsymbol{B}_r^{k+1} = \left( \sum_{t=1}^{T} \left( \boldsymbol{m}_{t,r}^k - \boldsymbol{A}_r^{k+1} \boldsymbol{m}_{t-1}^k \right) \boldsymbol{u}_t' \right) \left( \sum_{t=1}^{T} \boldsymbol{u}_t \boldsymbol{u}_t' + \boldsymbol{R}_{rr}^k \overline{\boldsymbol{\Gamma}}_{b,r}^{kd} \right)^{-1}, \tag{21}$$

$$\boldsymbol{C}_r^{k+1} = \left( \sum_{t=1}^{T} \left( \boldsymbol{y}_{t,r} - \boldsymbol{D}_r^k \boldsymbol{u}_t \right) \left( \boldsymbol{m}_t^k \right)' \right) \left( \sum_{t=1}^{T} \left( \boldsymbol{P}_t^k + \boldsymbol{m}_t^k \left( \boldsymbol{m}_t^k \right)' \right) + \boldsymbol{Q}_{rr}^k \overline{\boldsymbol{\Gamma}}_{c,r}^{kd} \right)^{-1}, \tag{22}$$

$$\boldsymbol{D}_r^{k+1} = \left( \sum_{t=1}^{T} \left( \boldsymbol{y}_{t,r} - \boldsymbol{C}_r^{k+1} \boldsymbol{m}_t^k \right) \boldsymbol{u}_t' \right) \left( \sum_{t=1}^{T} \boldsymbol{u}_t \boldsymbol{u}_t' + \boldsymbol{Q}_{rr}^k \overline{\boldsymbol{\Gamma}}_{d,r}^{kd} \right)^{-1}, \tag{23}$$

where $\boldsymbol{y}_{t,r}$ is the $r$th component of $\boldsymbol{y}_t$, $\boldsymbol{Q}_{rr}^k$ is the $rr$th component of $\boldsymbol{Q}^k$, and $\overline{\boldsymbol{\Gamma}}_{b,r}^{kd}$, $\overline{\boldsymbol{\Gamma}}_{c,r}^{kd}$, and $\overline{\boldsymbol{\Gamma}}_{d,r}^{kd}$ are defined as that of $\overline{\boldsymbol{\Gamma}}_{a,r}^{kd}$.

**Update procedures of $\boldsymbol{R}$ and $\boldsymbol{Q}$.** The derivative of $H\left( \boldsymbol{\Theta} \mid \boldsymbol{\Theta}^k \right)$ with respect to the $rr$th component of $\boldsymbol{R}$, denoted as $\boldsymbol{R}_{rr}$, is as follows (see Appendix B.3):

$$\frac{\partial H_1\left( \boldsymbol{A}^{k+1}, \boldsymbol{B}^{k+1}, \boldsymbol{R} \right)}{\partial \boldsymbol{R}_{rr}} = -\frac{T}{2\boldsymbol{R}_{rr}} + \frac{\sum_{t=1}^{T} \left( \boldsymbol{\pi}_{t,r}^k \right)^2 + \sum_{t=1}^{T} \boldsymbol{\Pi}_{t,rr}^k}{2\boldsymbol{R}_{rr}^2}, \tag{24}$$

where

$$\boldsymbol{\pi}_t^k = \boldsymbol{m}_t^k - \boldsymbol{A}^{k+1} \boldsymbol{m}_{t-1}^k - \boldsymbol{B}^{k+1} \boldsymbol{u}_t, \tag{25}$$

$$\boldsymbol{\Pi}_t^k = \boldsymbol{P}_t^k - \boldsymbol{A}^{k+1} \boldsymbol{P}_{t,t-1}^k - \boldsymbol{P}_{t,t-1}^k \left( \boldsymbol{A}^{k+1} \right)' + \boldsymbol{A}^{k+1} \boldsymbol{P}_{t-1}^k \left( \boldsymbol{A}^{k+1} \right)' \tag{26}$$

with $\boldsymbol{\pi}_{t,r}^k$ and $\boldsymbol{\Pi}_{t,rr}^k$ being the $r$th and $rr$th components of $\boldsymbol{\pi}_t^k$ and $\boldsymbol{\Pi}_t^k$, respectively. Setting equation 24 to zero yields the following update rule for $\boldsymbol{R}_{rr}$ at the $k$th iteration:

$$\boldsymbol{R}_{rr}^{k+1} = \frac{\sum_{t=1}^{T} \left( \boldsymbol{\pi}_{t,r}^k \right)^2 + \sum_{t=1}^{T} \boldsymbol{\Pi}_{t,rr}^k}{T}. \tag{27}$$

Similarly, we can update the $rr$th component of $\boldsymbol{Q}$, denoted as $\boldsymbol{Q}_{rr}$, at the $k$th iteration as follows:

$$\boldsymbol{Q}_{rr}^{k+1} = \frac{\sum_{t=1}^{T} \left( \boldsymbol{\psi}_{t,r}^k \right)^2 + \sum_{t=1}^{T} \boldsymbol{\Psi}_{t,rr}^k}{T}, \tag{28}$$

where

$$\boldsymbol{\psi}_t^k = \boldsymbol{y}_t - \boldsymbol{C}^{k+1} \boldsymbol{m}_t^k - \boldsymbol{D}^{k+1} \boldsymbol{u}_t, \quad \boldsymbol{\Psi}_t^k = \boldsymbol{C}^{k+1} \boldsymbol{P}_t^k \left( \boldsymbol{C}^{k+1} \right)' \tag{29}$$

with $\boldsymbol{\psi}_{t,r}^k$ and $\boldsymbol{\Psi}_{t,rr}^k$ being the $r$th and $rr$th components of $\boldsymbol{\psi}_t^k$ and $\boldsymbol{\Psi}_t^k$, respectively.

**Update procedures of $\boldsymbol{\Gamma}_a$, $\boldsymbol{\Gamma}_b$, $\boldsymbol{\Gamma}_c$, and $\boldsymbol{\Gamma}_d$.** Because each component of $\boldsymbol{\Gamma}_a$, $\boldsymbol{\Gamma}_b$, $\boldsymbol{\Gamma}_c$, and $\boldsymbol{\Gamma}_d$ is independent of the others, we can update them individually. For example, we can calculate the derivative of $H\left( \boldsymbol{\Theta} \mid \boldsymbol{\Theta}^k \right)$ with respect to $\boldsymbol{\Gamma}_{a,ij}$ at the $k$th iteration as follows:

$$\frac{\partial H_3\left( \boldsymbol{A}^{k+1}, \boldsymbol{B}^{k+1}, \boldsymbol{C}^{k+1}, \boldsymbol{D}^{k+1}, \boldsymbol{\Gamma}_a, \boldsymbol{\Gamma}_b^k, \boldsymbol{\Gamma}_c^k, \boldsymbol{\Gamma}_d^k \right)}{\partial \boldsymbol{\Gamma}_{a,ij}} = -\frac{2a_0 + 3}{2\boldsymbol{\Gamma}_{a,ij}} + \frac{\left( \boldsymbol{A}_{ij}^{k+1} \right)^2 + 2b_0}{2\boldsymbol{\Gamma}_{a,ij}^2}. \tag{30}$$

Setting equation 30 to zero and solving for $\boldsymbol{\Gamma}_{a,ij}$ leads to:

$$\boldsymbol{\Gamma}_{a,ij}^{k+1} = \frac{\left( \boldsymbol{A}_{ij}^{k+1} \right)^2 + 2b_0}{2a_0 + 3}. \tag{31}$$

Similarly, we can update each component of $\boldsymbol{\Gamma}_b$, $\boldsymbol{\Gamma}_c$, and $\boldsymbol{\Gamma}_d$ at the $k$th iteration as follows:

$$\boldsymbol{\Gamma}_{b,ij}^{k+1} = \frac{\left( \boldsymbol{B}_{ij}^{k+1} \right)^2 + 2b_0}{2a_0 + 3}, \quad \boldsymbol{\Gamma}_{c,ij}^{k+1} = \frac{\left( \boldsymbol{C}_{ij}^{k+1} \right)^2 + 2b_0}{2a_0 + 3}, \quad \boldsymbol{\Gamma}_{d,ij}^{k+1} = \frac{\left( \boldsymbol{D}_{ij}^{k+1} \right)^2 + 2b_0}{2a_0 + 3}. \tag{32}$$

Finally, we summarize the procedure for learning LSSMs with sparse system matrices in Appendix C.

### 3.5 GLOBAL CONVERGENCE ANALYSIS

Given $\boldsymbol{\Theta}^k$, the proposed learning algorithm presents the analytical mathematical expressions to derive $\boldsymbol{\Theta}^{k+1}$. Hence, we can define the proposed algorithm as a point-to-point mapping $\mathcal{A}(\cdot)$. Leveraging the Global Convergence Theorem (Luenberger et al., 1984), we can demonstrate that the proposed algorithm is globally convergent.

**Theorem 3.3.** *From any valid initialization point $\boldsymbol{\Theta}^0$, the limit point of the sequence $\{\boldsymbol{\Theta}^k\}_{k=1}^{\infty}$ generated via $\boldsymbol{\Theta}^{k+1} = \mathcal{A}\left(\boldsymbol{\Theta}^k\right)$ is a local maximum (or saddle point) of equation 6.*

*Proof.* See Appendix D. □

## 4 SIMILARITY TRANSFORMATION OF LSSMs

The similarity transformation provides an equivalent realization of original LSSMs by transforming states into different coordinate systems. For LSSMs, it is an important mathematical operation to analyze system properties like controllability, observability, and stability. Specifically, we can transform the state vector $\boldsymbol{x}_t$ into a new state vector $\overline{\boldsymbol{x}}_t$ through the relation $\overline{\boldsymbol{x}}_t = \boldsymbol{\Phi}\boldsymbol{x}_t$, where $\boldsymbol{\Phi} \in \mathbb{R}^{n \times n}$ is a nonsingular matrix. As such, we can derive an equivalent realization of the original LSSMs as follows (see Appendix E):

$$\overline{\boldsymbol{x}}_t = \overline{\boldsymbol{A}}\overline{\boldsymbol{x}}_{t-1} + \overline{\boldsymbol{B}}\boldsymbol{u}_t + \overline{\boldsymbol{\varepsilon}}_t, \tag{33}$$

$$\boldsymbol{y}_t = \overline{\boldsymbol{C}}\overline{\boldsymbol{x}}_t + \boldsymbol{D}\boldsymbol{u}_t + \boldsymbol{\omega}_t, \tag{34}$$

where

$$\overline{\boldsymbol{A}} = \boldsymbol{\Phi}\boldsymbol{A}\boldsymbol{\Phi}^{-1}, \quad \overline{\boldsymbol{B}} = \boldsymbol{\Phi}\boldsymbol{B}, \quad \overline{\boldsymbol{C}} = \boldsymbol{C}\boldsymbol{\Phi}^{-1}, \tag{35}$$

and $\overline{\boldsymbol{\varepsilon}}_t \sim \mathcal{N}\left(\boldsymbol{0}, \boldsymbol{\Phi}\boldsymbol{R}\boldsymbol{\Phi}'\right)$. However, the similarity transformation makes it particularly difficult to accurately learn system matrices. Given the input signals $\{\boldsymbol{u}_t\}_{t=1}^T$, the transformed LSSMs can produce the same output data $\{\boldsymbol{y}_t\}_{t=1}^T$ as the original LSSMs. Hence, classical learning algorithms for LSSMs only learn the system matrices up to a similar transformation (Viberg, 1994). For LSSMs with sparse system matrices, such a transformation changes not only the numerical values but, more importantly, the topological structure of the system matrices, resulting in misinterpretation of intrinsic working mechanisms.

### 4.1 BENEFIT OF SPARSITY-PROMOTING PRIORS

Unlike classical learning algorithms, the proposed algorithm learns LSSMs with sparse system matrices by adopting a sparsity-promoting prior to balance model complexity and modeling error. Given the sparsity constraint of system matrices, the similarity transformation cannot be applied using any arbitrary nonsingular matrix $\boldsymbol{\Phi}$. For the identifiable LSSMs, the nonsingular matrix is restricted to be a generalized permutation matrix; otherwise, the transformed LSSMs will include redundant parameters to describe the systems. For example, consider the LSSMs with sparse system matrices as follows:

$$\boldsymbol{A} = \begin{bmatrix} 0 & 0 & 0.8 \\ 0.8 & 0 & 0 \\ 0 & 0.8 & 0 \end{bmatrix}, \quad \boldsymbol{B} = \begin{bmatrix} 2 & 0 & 0 \\ 0 & 2 & 0 \\ 0 & 0 & 2 \end{bmatrix}, \quad \boldsymbol{C} = \begin{bmatrix} 2 & 0 & 0 \\ 0 & 0 & 2 \\ 0 & 2 & 0 \end{bmatrix}, \quad \boldsymbol{D} = \begin{bmatrix} 0 & 2 & 0 \\ 0 & 0 & 2 \\ 2 & 0 & 0 \end{bmatrix}. \tag{36}$$

It is easy to verify that such a system is identifiable, as the rank of system matrices is equal to the number of nonzero components. Hence, we can derive that the nonsingular matrix $\boldsymbol{\Phi}$ must be one of the following generalized permutation matrices:

$$\begin{bmatrix} a & 0 & 0 \\ 0 & b & 0 \\ 0 & 0 & c \end{bmatrix}, \quad \begin{bmatrix} a & 0 & 0 \\ 0 & 0 & b \\ 0 & c & 0 \end{bmatrix}, \quad \begin{bmatrix} 0 & a & 0 \\ b & 0 & 0 \\ 0 & 0 & c \end{bmatrix}, \quad \begin{bmatrix} 0 & a & 0 \\ 0 & 0 & b \\ c & 0 & 0 \end{bmatrix}, \quad \begin{bmatrix} 0 & 0 & a \\ b & 0 & 0 \\ 0 & c & 0 \end{bmatrix}, \quad \begin{bmatrix} 0 & 0 & a \\ 0 & b & 0 \\ c & 0 & 0 \end{bmatrix}, \tag{37}$$

where $a$, $b$, and $c$ are arbitrary constants. As such, the transformed system matrices $\overline{\boldsymbol{A}}, \overline{\boldsymbol{B}}$, and $\overline{\boldsymbol{C}}$ do not introduce additional parameters to increase model complexity.

Note that applying the similarity transformation with a generalized permutation matrix to the original state variables will scale their magnitudes and reorder them. However, it will scale the nonzero

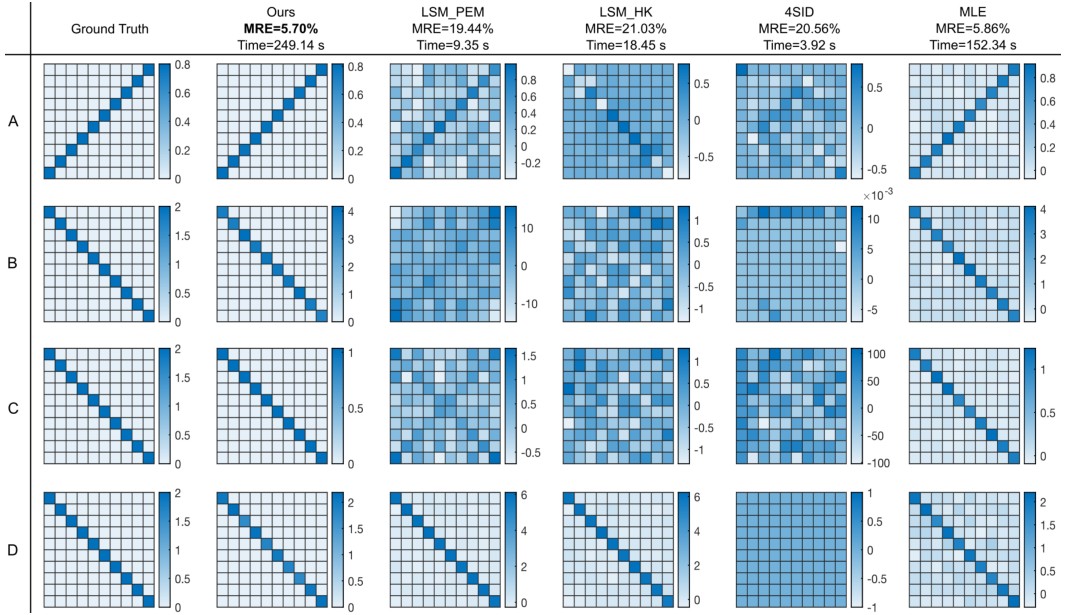

Figure 1: Experimental results of all the algorithms on the 10-dimensional synthetic system. Compared to the ground truth, only the proposed algorithm preserves the topological structure among the variables with $\mathbf{\Phi} \approx 2\boldsymbol{I}_{10}$. In addition, the proposed algorithm obtains the lowest MRE among all the algorithms. Experimental results on the more complex **non-diagonal** and **non-invertible** cases can be found in Appendix I.1 and Appendix I.2, respectively.

components and permute the rows or columns of system matrices accordingly. Hence, an additional advantage of the sparsity-promoting prior is its ability to preserve the inherent topological structure among the variables. While the learned system matrices differ from the true ones in numerical values, such a difference is only caused by the scaled definition of state variables, rather than a failure to capture the underlying system dynamics.

## 5 EXPERIMENT

In this section, we validate the proposed algorithm on simulation and real-world datasets. In addition, we compare the proposed algorithm with classical ones mentioned previously to demonstrate its superior performance, including LSM_PEM, LSM_HK, 4SID, and MLE. In all experiments, the dataset is split into training and testing sets with a 2:1 ratio, where 66.7% of the data is used for training and 33.3% for testing. Due to the lack of ground truth for real-world datasets, we use the mean relative error (MRE) to evaluate the performance of all the algorithms defined as follows: $\text{MRE} = \sum_{t=1}^{T} \frac{\|\boldsymbol{y}_t - \hat{\boldsymbol{y}}_t\|_2^2}{T\|\boldsymbol{y}_t\|_2^2}$, where $\{\hat{\boldsymbol{y}}_t\}_{t=1}^{T}$ is the sequence of data points generated by the learned LSSMs in response to the same input signals. The experiments are conducted using MATLAB 2022b on the PC with an Apple M1 Pro chip with 10-core CPU and 32GB RAM. Experimental results on simulation and real-world problems illustrate that the proposed algorithm can preserve the inherent topological structure among variables and significantly improve prediction accuracy over the classical ones. However, because the closed-form updates entail the inversion of a large number of matrices, the proposed algorithm involves high computational complexity (see Appendix F).

### 5.1 SYNTHETIC SYSTEMS

First, we test all the algorithms on a 10-dimensional synthetic system, indicating that each system matrix has 100 unknown parameters. Specifically, we consider $\boldsymbol{A}$ to be an anti-diagonal matrix with nonzero components equal to 0.8, and set $\boldsymbol{B} = \boldsymbol{C} = \boldsymbol{D} = 2\boldsymbol{I}_{10}$ and $\boldsymbol{R} = \boldsymbol{Q} = 0.81\boldsymbol{I}_{10}$. Because these system matrices are extremely sparse, accurately learning their topological structures is partic-

Table 1: Comparison results on the real-world industrial process systems

| Dataset | Ours | LSM_PEM | LSM_HK | 4SID | MLE |
|---|---|---|---|---|---|
| Evaporation system | **14.93%** (249.14 s) | 17.90% (9.35 s) | Inf (18.45 s) | 43.77% (3.92 s) | 20.14% (152.34 s) |
| Glass furnace | **23.63%** (45.62 s) | 62.62% (0.52 s) | 31.21% (6.20 s) | 24.32% (0.29 s) | 30.21% (33.36 s) |
| Steam generator | **20.70%** (441.88 s) | 22.70% (4.94 s) | 39.80% (84.52 s) | 29.26% (0.58 s) | 22.23% (299.10 s) |

ularly challenging. To generate data points, the value of $x_0$ is drawn from the Gaussian distribution with the mean diag$[I_{10}]$ and an identity covariance matrix, and the input signal $u_t$ is drawn from the uniform distribution on $[0, 2]$. As for algorithm implementation, we collect 2,100 data points and set the initial value of $A$, $B$, $C$, $D$, $R$, and $Q$ both to be $I_{10}$. Because the iterative numerical optimization method rarely yields exact zeros, the learned parameters below the predefined threshold are truncated to zero to accelerate convergence. The threshold selection procedure can be found in Appendix G. To ensure a fair comparison, the learned parameters of all the other algorithms that fall below this threshold are likewise set to zero.

Because the system matrices of the synthetic system are known, we can compare the learned system matrices with real ones directly. Figure 1 shows the learned system matrices of all the algorithms, together with the associated MRE and running time. Obviously, the learned system matrices of classical algorithms are completely different with the original ones in both numerical values and topological structures due to the similarity transformation, making it difficult for us to understand the system. However, sparsity-promoting priors will restrict the nonsingular matrix $\Phi$ of the similarity transformation to be a generalized permutation matrix for this system. By comparing the learned $B$ and $C$ of the proposed algorithm with real ones, we can derive $\Phi \approx 2I_{10}$, which is indeed consistent with the analysis in Section 4.1. Hence, the learned system matrices of the proposed algorithm preserve the inherent topological structure among the variables, differing only in numerical values due to the scaled definition of state variables.

We test the proposed algorithm on different initial values to demonstrate its robustness in Appendix H.1. We also conduct 20 independent trials to report the success rates of all the algorithms in learning the topological structure of the system matrices and the average MRE in Appendix H.2; experimental results illustrate that the proposed algorithm is stable across multiple runs.

In addition, we further demonstrate the effectiveness of the proposed algorithm on the **non-diagonal** and **non-invertible** cases in Appendix I.1 and Appendix I.2, respectively.

## 5.2 INDUSTRIAL PROCESS SYSTEMS

Next, we validate the proposed algorithm on the real-world datasets obtained from the Database for the Identification of Systems, which are standard datasets used for learning LSSMs (Zhu et al., 1994; Martens, 2010). While underlying physical systems may be non-linear, the learned LSSMs can provide a linear approximation of systems, enabling tractable analysis and control. To empirically ensure the performance of all the algorithms, we prefer to select datasets that contain at least 1,000 sample points and have multi-dimensional inputs and outputs.

**Evaporation System.** In industry, multiple-stage evaporators are widely used to reduce the water content of a product such as milk. The dataset is composed of 3-dimensional time-series with a length of 6,305. The inputs consist of the feed flow, vapor flow to the first evaporator stage, and cooling water flow to the condenser, while the outputs include the dry matter content, flow rate, and temperature of the product.

**Glass Furnace.** The second dataset comes from the Philips glass furnace, which is used to melt raw materials into glass. The glass furnace has two burners and one ventilator. Hence, the dataset includes two heating inputs and one cooling input with a length of 1,247. In addition, we collect three outputs from temperature sensors in a cross section of the furnace.

**Steam Generator.** The dataset comes from a boiler at Abbott Power Plant in Champaign IL, which is a dual-fuel (oil and gas) combustion unit used for both heating and electricity generation. It consists of four inputs (i.e., air flow rate, fuel flow rate, feedwater flow rate, and disturbance) and four outputs (i.e., drum pressure, excess oxygen, drum water level, and steam flow) with a length of 9,600.

Table 1 displays the MRE between the predicted outputs of all the learned LSSMs and the real one, and records the running time of each algorithm. Due to the lack of ground truth, the learned system matrices of all the algorithms are not depicted for comparison. Because the proposed algorithm obtains minimum MRE across all the datasets, experimental results demonstrate its superiority over the classical ones in real-world applications. Note that the hidden states can be defined in different coordinate systems as discussed in Section 4. As a result, the learned system matrices of classical algorithms may differ from the ground truth in both numerical values and topological structures. However, the proposed algorithm learns the system matrices by balancing model complexity and modeling error. Unlike classical algorithms, it will thus learn LSSMs with only the minimally required parameters to explain the time-series data. In particular, if the real-world systems are identifiable, the proposed algorithm preserves the topological structure among variables, which is more valuable for exploring interaction laws of systems.

## 6 DISCUSSION

To learn the LSSMs with sparse system matrices, we impose sparsity-promoting priors on system matrices to balance model complexity and modeling error. Following the MAP principle, we then learn system matrices by exploring the EM algorithm to maximize the joint posterior distribution composed of the priors and marginal likelihood function. Based on the Global Convergence Theorem, we demonstrate that the sequence generated by the proposed algorithm converges to a local maximum or saddle point of the posterior distribution. In addition, we explain why the sparsity-promoting prior is capable of retaining the inherent topological structure of LSSMs, as the non-singular matrix of the similarity transformation is limited to be a generalized permutation matrix. Hence, the proposed algorithm is more useful for us to explore the interaction laws of LSSMs compared to the classical ones.

There still remain some potential limitations associated with the proposed algorithm. First, the similarity transformation may shrink many parameters in system matrices to very small values, potentially leading to numerical errors. However, note that it is a common issue faced by all the learning algorithms for LSSMs. The other limitation is that the proposed algorithm is hard to deal with large-scale problems currently due to its high computational requirements. Hence, our future work will focus on reducing computation time to make the proposed algorithm applicable to large-scale settings. For instance, we can adopt the stochastic EM algorithm to reduce the computational cost by using a mini-batch of data instead of the full batch during the expectation step (Chen et al., 2018). In addition, we hope to explore how to ensure that the learned system matrices are exactly the same as the true ones by imposing additional constraints on system matrices beyond the sparsity constraint. Overall, we believe that the proposed algorithm sheds light on the learning of LSSMs with sparse system matrices.

ACKNOWLEDGMENTS

This work is supported by the National Natural Science Foundation of China (62503185, 62303118) and Doctoral Foundation of Fuyang Normal University (2021KYQD0034).

ETHICS STATEMENT

This paper presents theoretical research whose goal is to advance the field of Machine Learning. Because it does not involve human participants, animals, sensitive personal data, or other foreseeable ethical concerns outlined in the ICLR Code of Ethics, no specific ethical issues arise from this paper.

REPRODUCIBILITY STATEMENT

To ensure the reproducibility of this research, the main text and Appendix provide detailed theoretical derivations and proofs of the proposed algorithm. Source codes are available on GitHub at `https://github.com/ArthinYS/Learning-Sparse-LSSM`. Moreover, the real-world datasets used in our experiments are publicly available at `https://homes.esat.kuleuven.be/~smc/daisy/`.

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

# A    SPARSITY-PROMOTING PRIOR

Besides $\boldsymbol{A}$, we also impose the sparsity-promoting priors on $\boldsymbol{B}$, $\boldsymbol{C}$, and $\boldsymbol{D}$ as follows:

$$p\left(\boldsymbol{B} \mid \boldsymbol{\Gamma}_b\right) = \prod_{i=1}^{n}\prod_{j=1}^{p} p\left(\boldsymbol{B}_{ij} \mid \boldsymbol{\Gamma}_{b,ij}\right) = \prod_{i=1}^{n}\prod_{j=1}^{p} \frac{1}{\sqrt{2\pi\boldsymbol{\Gamma}_{b,ij}}} \exp\left(-\frac{\boldsymbol{B}_{ij}^2}{2\boldsymbol{\Gamma}_{b,ij}}\right), \tag{38}$$

$$p\left(\boldsymbol{C} \mid \boldsymbol{\Gamma}_c\right) = \prod_{i=1}^{m}\prod_{j=1}^{n} p\left(\boldsymbol{C}_{ij} \mid \boldsymbol{\Gamma}_{c,ij}\right) = \prod_{i=1}^{m}\prod_{j=1}^{n} \frac{1}{\sqrt{2\pi\boldsymbol{\Gamma}_{c,ij}}} \exp\left(-\frac{\boldsymbol{C}_{ij}^2}{2\boldsymbol{\Gamma}_{c,ij}}\right), \tag{39}$$

$$p\left(\boldsymbol{D} \mid \boldsymbol{\Gamma}_d\right) = \prod_{i=1}^{m}\prod_{j=1}^{p} p\left(\boldsymbol{D}_{ij} \mid \boldsymbol{\Gamma}_{d,ij}\right) = \prod_{i=1}^{m}\prod_{j=1}^{p} \frac{1}{\sqrt{2\pi\boldsymbol{\Gamma}_{d,ij}}} \exp\left(-\frac{\boldsymbol{D}_{ij}^2}{2\boldsymbol{\Gamma}_{d,ij}}\right), \tag{40}$$

where $\boldsymbol{\Gamma}_{b,ij}, \boldsymbol{\Gamma}_{c,ij}$, and $\boldsymbol{\Gamma}_{d,ij}$ are the $ij$th components of $\boldsymbol{\Gamma}_b, \boldsymbol{\Gamma}_c$, and $\boldsymbol{\Gamma}_d$, respectively. To complete the hierarchy, the Inverse-Gamma distribution prior is imposed on each component of $\boldsymbol{\Gamma}_b, \boldsymbol{\Gamma}_c$, and $\boldsymbol{\Gamma}_d$ as follows:

$$p(\boldsymbol{\Gamma}_b) = \prod_{i=1}^{n}\prod_{j=1}^{p} \frac{a_0^{b_0}}{\Gamma(a_0)} \boldsymbol{\Gamma}_{b,ij}^{-a_0-1} \exp\left(-\frac{b_0}{\boldsymbol{\Gamma}_{b,ij}}\right), \tag{41}$$

$$p(\boldsymbol{\Gamma}_c) = \prod_{i=1}^{m}\prod_{j=1}^{n} \frac{a_0^{b_0}}{\Gamma(a_0)} \boldsymbol{\Gamma}_{c,ij}^{-a_0-1} \exp\left(-\frac{b_0}{\boldsymbol{\Gamma}_{c,ij}}\right), \tag{42}$$

$$p(\boldsymbol{\Gamma}_d) = \prod_{i=1}^{m}\prod_{j=1}^{p} \frac{a_0^{b_0}}{\Gamma(a_0)} \boldsymbol{\Gamma}_{d,ij}^{-a_0-1} \exp\left(-\frac{b_0}{\boldsymbol{\Gamma}_{d,ij}}\right). \tag{43}$$

# B    DETAILED MATHEMATICAL DERIVATION

## B.1    DERIVATION OF EQUATION 15

Given the conditional independence between the variables, we can derive

$$
\begin{aligned}
&H\left(\boldsymbol{\Theta} \mid \boldsymbol{\Theta}^k\right) \\
&= \mathbb{E}_{\boldsymbol{X}^k}\left[\log\left(p(\boldsymbol{Y}, \boldsymbol{X} \mid \boldsymbol{\Theta})p(\boldsymbol{\Theta})\right)\right] \\
&= \mathbb{E}_{\boldsymbol{X}^k}\left[\log\left(p(\boldsymbol{Y} \mid \boldsymbol{X}, \boldsymbol{\Theta})p(\boldsymbol{X} \mid \boldsymbol{\Theta})p(\boldsymbol{\Theta})\right)\right] \\
&= \mathbb{E}_{\boldsymbol{X}^k}\left[\log\left(p(\boldsymbol{Y} \mid \boldsymbol{X}, \boldsymbol{C}, \boldsymbol{D}, \boldsymbol{Q})p(\boldsymbol{X} \mid \boldsymbol{A}, \boldsymbol{B}, \boldsymbol{R})p(\boldsymbol{A}, \boldsymbol{\Gamma}_a, \boldsymbol{B}, \boldsymbol{\Gamma}_b, \boldsymbol{C}, \boldsymbol{\Gamma}_c, \boldsymbol{D}, \boldsymbol{\Gamma}_d)\right)\right] \\
&= \mathbb{E}_{\boldsymbol{X}^k}\left[\log\left(p(\boldsymbol{Y} \mid \boldsymbol{X}, \boldsymbol{C}, \boldsymbol{D}, \boldsymbol{Q})p(\boldsymbol{X} \mid \boldsymbol{A}, \boldsymbol{B}, \boldsymbol{R})p(\boldsymbol{A}, \boldsymbol{\Gamma}_a)p(\boldsymbol{B}, \boldsymbol{\Gamma}_b)p(\boldsymbol{C}, \boldsymbol{\Gamma}_c)p(\boldsymbol{D}, \boldsymbol{\Gamma}_d)\right)\right] \\
&= \underbrace{\mathbb{E}_{\boldsymbol{X}^k}\left[\log p(\boldsymbol{X} \mid \boldsymbol{A}, \boldsymbol{B}, \boldsymbol{R})\right]}_{H_1(\boldsymbol{A}, \boldsymbol{B}, \boldsymbol{R})} + \underbrace{\mathbb{E}_{\boldsymbol{X}^k}\left[\log p(\boldsymbol{Y} \mid \boldsymbol{X}, \boldsymbol{C}, \boldsymbol{D}, \boldsymbol{Q})\right]}_{H_2(\boldsymbol{C}, \boldsymbol{D}, \boldsymbol{Q})} \\
&+ \underbrace{\mathbb{E}_{\boldsymbol{X}^k}\left[\log\left(p(\boldsymbol{A}, \boldsymbol{\Gamma}_a)p(\boldsymbol{B}, \boldsymbol{\Gamma}_b)p(\boldsymbol{C}, \boldsymbol{\Gamma}_c)p(\boldsymbol{D}, \boldsymbol{\Gamma}_d)\right)\right]}_{H_3(\boldsymbol{A}, \boldsymbol{B}, \boldsymbol{C}, \boldsymbol{D}, \boldsymbol{\Gamma}_a, \boldsymbol{\Gamma}_b, \boldsymbol{\Gamma}_c, \boldsymbol{\Gamma}_d)} .
\end{aligned}
\tag{44}
$$

**Explicit mathematical expression of $H_1(\boldsymbol{A}, \boldsymbol{B}, \boldsymbol{R})$.** Based on equation 1 and the chain rule in probability, we can derive

$$
\begin{aligned}
&p(\boldsymbol{X} \mid \boldsymbol{A}, \boldsymbol{B}, \boldsymbol{R}) \\
&= p(\boldsymbol{x}_0)\prod_{t=1}^{T} p(\boldsymbol{x}_t \mid \boldsymbol{x}_{t-1}, \boldsymbol{A}, \boldsymbol{B}, \boldsymbol{R}) \\
&\propto \prod_{t=1}^{T} \mid \boldsymbol{R} \mid^{-\frac{1}{2}} \exp\left(-\frac{(\boldsymbol{x}_t - \boldsymbol{A}\boldsymbol{x}_{t-1} - \boldsymbol{B}\boldsymbol{u}_t)'\boldsymbol{R}^{-1}(\boldsymbol{x}_t - \boldsymbol{A}\boldsymbol{x}_{t-1} - \boldsymbol{B}\boldsymbol{u}_t)}{2}\right) .
\end{aligned}
\tag{45}
$$

Hence,

$$
\begin{aligned}
&H_1(\boldsymbol{A}, \boldsymbol{B}, \boldsymbol{R}) \\
&= \mathbb{E}_{\boldsymbol{X}^k}\left[\log p(\boldsymbol{X} \mid \boldsymbol{A}, \boldsymbol{B}, \boldsymbol{R})\right] \\
&= \mathbb{E}_{\boldsymbol{X}^k}\left[-\frac{T\log \mid \boldsymbol{R} \mid + \sum_{t=1}^{T}(\boldsymbol{x}_t - \boldsymbol{A}\boldsymbol{x}_{t-1} - \boldsymbol{B}\boldsymbol{u}_t)'\boldsymbol{R}^{-1}(\boldsymbol{x}_t - \boldsymbol{A}\boldsymbol{x}_{t-1} - \boldsymbol{B}\boldsymbol{u}_t)}{2}\right] \\
&= -\frac{T\log \mid \boldsymbol{R} \mid + \sum_{t=1}^{T}\mathbb{E}_{\boldsymbol{X}^k}\left[(\boldsymbol{x}_t - \boldsymbol{A}\boldsymbol{x}_{t-1} - \boldsymbol{B}\boldsymbol{u}_t)'\boldsymbol{R}^{-1}(\boldsymbol{x}_t - \boldsymbol{A}\boldsymbol{x}_{t-1} - \boldsymbol{B}\boldsymbol{u}_t)\right]}{2} \\
&= -\frac{T\log \mid \boldsymbol{R} \mid + \sum_{t=1}^{T}\left(\boldsymbol{m}_t^k - \boldsymbol{A}\boldsymbol{m}_{t-1}^k - \boldsymbol{B}\boldsymbol{u}_t\right)'\boldsymbol{R}^{-1}\left(\boldsymbol{m}_t^k - \boldsymbol{A}\boldsymbol{m}_{t-1}^k - \boldsymbol{B}\boldsymbol{u}_t\right)}{2} \\
&\quad - \frac{\sum_{t=1}^{T}\left(\operatorname{Tr}\left(\boldsymbol{R}^{-1}\boldsymbol{P}_t^k\right) - \operatorname{Tr}\left(\boldsymbol{R}^{-1}\boldsymbol{A}\boldsymbol{P}_{t,t-1}^k\right) - \operatorname{Tr}\left(\boldsymbol{A}'\boldsymbol{R}^{-1}\boldsymbol{P}_{t,t-1}^k\right) + \operatorname{Tr}\left(\boldsymbol{A}'\boldsymbol{R}^{-1}\boldsymbol{A}\boldsymbol{P}_{t-1}^k\right)\right)}{2} .
\end{aligned}
\tag{46}
$$

**Explicit mathematical expression of $H_2(\boldsymbol{C}, \boldsymbol{D}, \boldsymbol{Q})$.** Based on equation 2, we can derive

$$
\begin{aligned}
&p(\boldsymbol{Y} \mid \boldsymbol{X}, \boldsymbol{C}, \boldsymbol{D}, \boldsymbol{Q}) \\
&= \prod_{t=1}^{T} p(\boldsymbol{y}_t \mid \boldsymbol{x}_t, \boldsymbol{C}, \boldsymbol{D}) \\
&\propto \prod_{t=1}^{T} \mid \boldsymbol{Q} \mid^{-\frac{1}{2}} \exp\left(-\frac{(\boldsymbol{y}_t - \boldsymbol{C}\boldsymbol{x}_t - \boldsymbol{D}\boldsymbol{u}_t)'\boldsymbol{Q}^{-1}(\boldsymbol{y}_t - \boldsymbol{C}\boldsymbol{x}_t - \boldsymbol{D}\boldsymbol{u}_t)}{2}\right) .
\end{aligned}
\tag{47}
$$

Hence,

$$
\begin{aligned}
& H_2(\boldsymbol{C}, \boldsymbol{D}, \boldsymbol{Q}) \\
&= \mathbb{E}_{\boldsymbol{X}^k}[\log p(\boldsymbol{Y} \mid \boldsymbol{X}, \boldsymbol{C}, \boldsymbol{D}, \boldsymbol{Q})] \\
&= -\frac{\mathbb{E}_{\boldsymbol{X}^k}\left[T \log \mid \boldsymbol{Q} \mid + \sum_{t=1}^{T}(\boldsymbol{y}_t - \boldsymbol{C}\boldsymbol{x}_t - \boldsymbol{D}\boldsymbol{u}_t)'\boldsymbol{Q}^{-1}(\boldsymbol{y}_t - \boldsymbol{C}\boldsymbol{x}_t - \boldsymbol{D}\boldsymbol{u}_t)\right]}{2} \\
&= -\frac{T \log \mid \boldsymbol{Q} \mid + \sum_{t=1}^{T} \mathbb{E}_{\boldsymbol{X}^k}\left[(\boldsymbol{y}_t - \boldsymbol{C}\boldsymbol{x}_t - \boldsymbol{D}\boldsymbol{u}_t)'\boldsymbol{Q}^{-1}(\boldsymbol{y}_t - \boldsymbol{C}\boldsymbol{x}_t - \boldsymbol{D}\boldsymbol{u}_t)\right]}{2} \\
&= -\frac{T \log \mid \boldsymbol{Q} \mid + \sum_{t=1}^{T}\left((\boldsymbol{y}_t - \boldsymbol{C}\boldsymbol{m}_t^k - \boldsymbol{D}\boldsymbol{u}_t)'\boldsymbol{Q}^{-1}(\boldsymbol{y}_t - \boldsymbol{C}\boldsymbol{m}_t^k - \boldsymbol{D}\boldsymbol{u}_t) + \operatorname{Tr}\left(\boldsymbol{C}'\boldsymbol{Q}^{-1}\boldsymbol{C}\boldsymbol{P}_t^k\right)\right)}{2}.
\end{aligned}
\tag{48}
$$

**Explicit mathematical expression of** $H_3(\boldsymbol{A}, \boldsymbol{B}, \boldsymbol{C}, \boldsymbol{D}, \boldsymbol{\Gamma}_a, \boldsymbol{\Gamma}_b, \boldsymbol{\Gamma}_c, \boldsymbol{\Gamma}_d)$**.** Based on the priors imposed on the system matrices and corresponding hyperparameters, we can derive:

$$
\begin{aligned}
& p(\boldsymbol{A}, \boldsymbol{\Gamma}_a)p(\boldsymbol{B}, \boldsymbol{\Gamma}_b)p(\boldsymbol{C}, \boldsymbol{\Gamma}_c)p(\boldsymbol{D}, \boldsymbol{\Gamma}_d) \\
&= p(\boldsymbol{A} \mid \boldsymbol{\Gamma}_a)p(\boldsymbol{\Gamma}_a)p(\boldsymbol{B} \mid \boldsymbol{\Gamma}_b)p(\boldsymbol{\Gamma}_b)p(\boldsymbol{C} \mid \boldsymbol{\Gamma}_c)p(\boldsymbol{\Gamma}_c)p(\boldsymbol{D} \mid \boldsymbol{\Gamma}_d)p(\boldsymbol{\Gamma}_d) \\
&\propto \prod_{i=1}^{n}\prod_{j=1}^{n} \boldsymbol{\Gamma}_{a,ij}^{-\frac{2a_0+3}{2}} \exp\left(-\frac{\boldsymbol{A}_{ij}^2 + 2b_0}{2\boldsymbol{\Gamma}_{a,ij}}\right) \times \prod_{i=1}^{n}\prod_{j=1}^{p} \boldsymbol{\Gamma}_{b,ij}^{-\frac{2a_0+3}{2}} \exp\left(-\frac{\boldsymbol{B}_{ij}^2 + 2b_0}{2\boldsymbol{\Gamma}_{b,ij}}\right) \\
&\times \prod_{i=1}^{m}\prod_{j=1}^{n} \boldsymbol{\Gamma}_{c,ij}^{-\frac{2a_0+3}{2}} \exp\left(-\frac{\boldsymbol{C}_{ij}^2 + 2b_0}{2\boldsymbol{\Gamma}_{c,ij}}\right) \times \prod_{i=1}^{m}\prod_{j=1}^{p} \boldsymbol{\Gamma}_{d,ij}^{-\frac{2a_0+3}{2}} \exp\left(-\frac{\boldsymbol{D}_{ij}^2 + 2b_0}{2\boldsymbol{\Gamma}_{d,ij}}\right).
\end{aligned}
\tag{49}
$$

Hence, we have

$$
\begin{aligned}
& H_3(\boldsymbol{A}, \boldsymbol{B}, \boldsymbol{C}, \boldsymbol{D}, \boldsymbol{\Gamma}_a, \boldsymbol{\Gamma}_b, \boldsymbol{\Gamma}_c, \boldsymbol{\Gamma}_d) \\
&= \mathbb{E}_{\boldsymbol{X}^k}\left[\log\left(p(\boldsymbol{A}, \boldsymbol{\Gamma}_a)p(\boldsymbol{B}, \boldsymbol{\Gamma}_b)p(\boldsymbol{C}, \boldsymbol{\Gamma}_c)p(\boldsymbol{D}, \boldsymbol{\Gamma}_d)\right)\right] \\
&= \mathbb{E}_{\boldsymbol{X}^k}\left[\log\left(p(\boldsymbol{A} \mid \boldsymbol{\Gamma}_a)p(\boldsymbol{\Gamma}_a)p(\boldsymbol{B} \mid \boldsymbol{\Gamma}_b)p(\boldsymbol{\Gamma}_b)p(\boldsymbol{C} \mid \boldsymbol{\Gamma}_c)p(\boldsymbol{\Gamma}_c)p(\boldsymbol{D} \mid \boldsymbol{\Gamma}_d)p(\boldsymbol{\Gamma}_d)\right)\right] \\
&= -\sum_{i=1}^{n}\sum_{j=1}^{n}\left(\frac{(2a_0+3)\log \mid \boldsymbol{\Gamma}_{a,ij} \mid}{2} + \frac{\boldsymbol{A}_{ij}^2 + 2b_0}{2\boldsymbol{\Gamma}_{a,ij}}\right) \\
& -\sum_{i=1}^{n}\sum_{j=1}^{p}\left(\frac{(2a_0+3)\log \mid \boldsymbol{\Gamma}_{b,ij} \mid}{2} + \frac{\boldsymbol{B}_{ij}^2 + 2b_0}{2\boldsymbol{\Gamma}_{b,ij}}\right) \\
& -\sum_{i=1}^{m}\sum_{j=1}^{n}\left(\frac{(2a_0+3)\log \mid \boldsymbol{\Gamma}_{c,ij} \mid}{2} + \frac{\boldsymbol{C}_{ij}^2 + 2b_0}{2\boldsymbol{\Gamma}_{c,ij}}\right) \\
& -\sum_{i=1}^{m}\sum_{j=1}^{p}\left(\frac{(2a_0+3)\log \mid \boldsymbol{\Gamma}_{d,ij} \mid}{2} + \frac{\boldsymbol{D}_{ij}^2 + 2b_0}{2\boldsymbol{\Gamma}_{d,ij}}\right).
\end{aligned}
\tag{50}
$$

## B.2 Derivation of equation 19

Because $H_1(\boldsymbol{A}, \boldsymbol{B}, \boldsymbol{R})$ and $H_3(\boldsymbol{A}, \boldsymbol{B}, \boldsymbol{C}, \boldsymbol{D}, \boldsymbol{\Gamma}_a, \boldsymbol{\Gamma}_b, \boldsymbol{\Gamma}_c, \boldsymbol{\Gamma}_d)$ involve $\boldsymbol{A}$, we can derive the term related to $\boldsymbol{A}_r$ at the $k$th iteration as follows:

$$
\begin{aligned}
& H_1\left(\boldsymbol{A}, \boldsymbol{B}^k, \boldsymbol{R}^k\right) + H_3\left(\boldsymbol{A}, \boldsymbol{B}^k, \boldsymbol{C}^k, \boldsymbol{D}^k, \boldsymbol{\Gamma}_a^k, \boldsymbol{\Gamma}_b^k, \boldsymbol{\Gamma}_c^k, \boldsymbol{\Gamma}_d^k\right) \\
&= -\frac{\sum_{t=1}^{T}\sum_{r=1}^{n}\left(\boldsymbol{R}_{rr}^k\right)^{-1}\left(\operatorname{Tr}\left(\boldsymbol{A}_r'\boldsymbol{A}_r\boldsymbol{P}_{t-1}^k\right) - 2\boldsymbol{A}_r\left(\boldsymbol{P}_{t,t-1,r}^k\right)'\right) + \sum_{r=1}^{n}\boldsymbol{A}_r\overline{\boldsymbol{\Gamma}}_{a,r}^{kd}\boldsymbol{A}_r'}{2} \\
& -\frac{\sum_{t=1}^{T}\sum_{r=1}^{n}\left(\boldsymbol{R}_{rr}^k\right)^{-1}\left(\boldsymbol{m}_{t,r}^k - \boldsymbol{A}_r\boldsymbol{m}_{t-1}^k - \boldsymbol{B}_r^k\boldsymbol{u}_t\right)\left(\boldsymbol{m}_{t,r}^k - \boldsymbol{A}_r\boldsymbol{m}_{t-1}^k - \boldsymbol{B}_r^k\boldsymbol{u}_t\right)'}{2} + c,
\end{aligned}
\tag{51}
$$

where $c$ is the term independent of $\boldsymbol{A}_r$. Hence, we can calculate the derivative of $H(\boldsymbol{\Theta} \mid \boldsymbol{\Theta}^k)$ with respect to $\boldsymbol{A}_r$ as follows:

$$
\begin{aligned}
&\frac{\partial H(\boldsymbol{\Theta} \mid \boldsymbol{\Theta}^k)}{\partial \boldsymbol{A}_r} \\
&= \frac{\partial H_1\left(\boldsymbol{A}, \boldsymbol{B}^k, \boldsymbol{R}^k\right)}{\partial \boldsymbol{A}_r} + \frac{\partial H_3\left(\boldsymbol{A}, \boldsymbol{B}^k, \boldsymbol{C}^k, \boldsymbol{D}^k, \boldsymbol{\Gamma}_a^k, \boldsymbol{\Gamma}_b^k, \boldsymbol{\Gamma}_c^k, \boldsymbol{\Gamma}_d^k\right)}{\partial \boldsymbol{A}_r} \\
&= \sum_{t=1}^{T} \left(\boldsymbol{R}_{rr}^k\right)^{-1} \left(\boldsymbol{P}_{t,t-1,r}^k + \left(\boldsymbol{m}_{t,r}^k - \boldsymbol{A}_r \boldsymbol{m}_{t-1}^k - \boldsymbol{B}_r^k \boldsymbol{u}_t\right)\left(\boldsymbol{m}_{t-1}^k\right)' - \boldsymbol{A}_r \boldsymbol{P}_{t-1}^k\right) - \boldsymbol{A}_r \overline{\boldsymbol{\Gamma}}_{a,r}^{kd}. \quad (52)
\end{aligned}
$$

Setting equation 52 to zero leads to

$$
\sum_{t=1}^{T} \boldsymbol{A}_r \left(\boldsymbol{m}_{t-1}^k \left(\boldsymbol{m}_{t-1}^k\right)' + \boldsymbol{P}_{t-1}^k\right) + \boldsymbol{A}_r \boldsymbol{R}_{rr}^k \overline{\boldsymbol{\Gamma}}_{a,r}^{kd} = \sum_{t=1}^{T} \left(\boldsymbol{P}_{t,t-1,r}^k + \left(\boldsymbol{m}_{t,r}^k - \boldsymbol{B}_r^k \boldsymbol{u}_t\right)\left(\boldsymbol{m}_{t-1}^k\right)'\right). \tag{53}
$$

Hence, we can update $\boldsymbol{A}_r$ at the $k$th iteration as follows:

$$
\begin{aligned}
\boldsymbol{A}_r^{k+1} = &\left(\sum_{t=1}^{T} \left(\left(\boldsymbol{m}_{t,r}^k - \boldsymbol{B}_r^k \boldsymbol{u}_t\right)\left(\boldsymbol{m}_{t-1}^k\right)' + \boldsymbol{P}_{t,t-1,r}^k\right)\right) \\
&\times \left(\sum_{t=1}^{T} \left(\boldsymbol{P}_{t-1}^k + \boldsymbol{m}_{t-1}^k \left(\boldsymbol{m}_{t-1}^k\right)'\right) + \boldsymbol{R}_{rr}^k \boldsymbol{\Gamma}_{a,r}^{kd}\right)^{-1}. \tag{54}
\end{aligned}
$$

### B.3 DERIVATION OF EQUATION 24

Given that $\boldsymbol{R}$ is a diagonal matrix, we can re-express $H_1\left(\boldsymbol{A}^{k+1}, \boldsymbol{B}^{k+1}, \boldsymbol{R}\right)$ as follows:

$$
\begin{aligned}
&H_1\left(\boldsymbol{A}^{k+1}, \boldsymbol{B}^{k+1}, \boldsymbol{R}\right) \\
&= -\frac{T \sum_{r=1}^{n} \log \boldsymbol{R}_{rr}}{2} - \frac{\sum_{r=1}^{n} \sum_{t=1}^{T} \boldsymbol{R}_{rr}^{-1} \left(\boldsymbol{\pi}_{t,r}^k\right)^2}{2} - \frac{\sum_{r=1}^{n} \sum_{t=1}^{T} \boldsymbol{R}_{rr}^{-1} \boldsymbol{\Pi}_{t,rr}^k}{2}, \tag{55}
\end{aligned}
$$

where

$$
\boldsymbol{\pi}_t^k = \boldsymbol{m}_t^k - \boldsymbol{A}^{k+1} \boldsymbol{m}_{t-1}^k - \boldsymbol{B}^{k+1} \boldsymbol{u}_t, \tag{56}
$$

$$
\boldsymbol{\Pi}_t^k = \boldsymbol{P}_t^k - \boldsymbol{A}^{k+1} \boldsymbol{P}_{t,t-1}^k - \boldsymbol{P}_{t,t-1}^k \left(\boldsymbol{A}^{k+1}\right)' + \boldsymbol{A}^{k+1} \boldsymbol{P}_{t-1}^k \left(\boldsymbol{A}^{k+1}\right)' \tag{57}
$$

with $\boldsymbol{\pi}_{t,r}^k$ and $\boldsymbol{\Pi}_{t,rr}^k$ being the $r$th and $rr$th components of $\boldsymbol{\pi}_t^k$ and $\boldsymbol{\Pi}_t^k$, respectively. Hence, we can calculate the derivative of $H(\boldsymbol{\Theta} \mid \boldsymbol{\Theta}^k)$ with respect to $\boldsymbol{R}_{rr}$ as follows:

$$
\frac{\partial H(\boldsymbol{\Theta} \mid \boldsymbol{\Theta}^k)}{\partial \boldsymbol{R}_{rr}} = \frac{\partial H_1\left(\boldsymbol{A}^{k+1}, \boldsymbol{B}^{k+1}, \boldsymbol{R}\right)}{\partial \boldsymbol{R}_{rr}} = -\frac{T}{2\boldsymbol{R}_{rr}} + \frac{\sum_{t=1}^{T}\left(\boldsymbol{\pi}_{t,r}^k\right)^2}{2\boldsymbol{R}_{rr}^2} + \frac{\sum_{t=1}^{T} \boldsymbol{\Pi}_{t,rr}^k}{2\boldsymbol{R}_{rr}^2}. \tag{58}
$$

Setting equation 58 to zero leads to the update rule for $\boldsymbol{R}_{rr}$ at the $k$th iteration as follows:

$$
\boldsymbol{R}_{rr}^{k+1} = \frac{\sum_{t=1}^{T}\left(\boldsymbol{\pi}_{t,r}^k\right)^2 + \sum_{t=1}^{T} \boldsymbol{\Pi}_{t,rr}^k}{T}. \tag{59}
$$

## C  PSEUDOCODE FOR LEARNING LSSMs WITH SPARSE SYSTEM MATRICES

---

**Algorithm 1** The proposed algorithm for learning LSSMs with sparse system matrices

---

**Input:** Time-series data $\{(\boldsymbol{u}_t, \boldsymbol{y}_t)\}_{t=1}^T$, initial guess of $\boldsymbol{\Theta}$, and maximum number of iterations $k_{max}$

**for** $k = 1$ **to** $k_{max}$ **do**

  **for** $t = 1$ **to** $T$ **do**

    Update the mean $\boldsymbol{m}_t^k$ and covariance $\boldsymbol{P}_t^k$ of $\boldsymbol{x}_t$ via equation 9 and equation 10, respectively

    Update the covariance $\boldsymbol{P}_{t,t-1}^k$ between $\boldsymbol{x}_t$ and $\boldsymbol{x}_{t-1}$ via equation 14

  **end for**

  Update system matrices $\boldsymbol{A}$, $\boldsymbol{B}$, $\boldsymbol{C}$, and $\boldsymbol{D}$ via equation 20–equation 23, respectively

  Update noise covariance matrices $\boldsymbol{R}$ and $\boldsymbol{Q}$ via equation 27 and equation 28, respectively

  Update hyperparameter matrices $\boldsymbol{\Gamma}_a$, $\boldsymbol{\Gamma}_b$, $\boldsymbol{\Gamma}_c$, and $\boldsymbol{\Gamma}_d$ via equation 31 and equation 32, respectively

  **if** a stopping criterion is satisfied **then**

    Break

  **end if**

**end for**

**Output:** System matrices $\boldsymbol{A}$, $\boldsymbol{B}$, $\boldsymbol{C}$, and $\boldsymbol{D}$, noise covariance matrices $\boldsymbol{R}$ and $\boldsymbol{Q}$

---

## D  PROOF OF THEOREM 3.3

*Proof.* To illustrate the global convergence of the proposed algorithm, we need to demonstrate that it satisfies the three necessary conditions required by the Global Convergence Theorem (Luenberger et al., 1984). **For $\mathcal{A}(\cdot)$ and $\boldsymbol{\Omega} = \{\boldsymbol{\Theta} : \nabla_{\boldsymbol{\Theta}} p(\boldsymbol{\Theta} \mid \boldsymbol{Y}) = 0\}$, $\mathcal{L}(\boldsymbol{\Theta}) = -p(\boldsymbol{\Theta} \mid \boldsymbol{Y})$ is a descent function.** For the point $\boldsymbol{\Theta}^k$ in $\boldsymbol{\Omega}$, it is straightforward to conclude that $\mathcal{L}(\boldsymbol{\Theta}^{k+1}) \leq \mathcal{L}(\boldsymbol{\Theta}^k)$ following the basic property of the EM algorithm. For the point $\boldsymbol{\Theta}^k$ outside $\boldsymbol{\Omega}$, as $H(\boldsymbol{\Theta} \mid \boldsymbol{\Theta}^k)$ is continuous in both arguments, we have $\mathcal{L}(\boldsymbol{\Theta}^{k+1}) < \mathcal{L}(\boldsymbol{\Theta}^k)$ (see Theorem 2 in Wu (1983)). Hence, $\mathcal{L}(\boldsymbol{\Theta})$ is a descent function for $\boldsymbol{\Omega}$ and $\mathcal{A}(\cdot)$. **The sequence $\{\boldsymbol{\Theta}^k\}_{k=1}^\infty$ is contained in a compact set.** If any component of $\boldsymbol{\Theta}$ is unbounded, $\mathcal{L}(\boldsymbol{\Theta})$ tends to infinity. Given $\mathcal{L}(\boldsymbol{\Theta}^{k+1}) \leq \mathcal{L}(\boldsymbol{\Theta}^k)$, there must exists a compact set containing the sequence $\{\boldsymbol{\Theta}^k\}_{k=1}^\infty$. **The mapping $\mathcal{A}(\cdot)$ is closed at points outside $\boldsymbol{\Omega}$.** Because the proposed algorithm updates $\boldsymbol{\Theta}^{k+1}$ from $\boldsymbol{\Theta}^k$ via analytical mathematical expressions, and all elementary functions involved are continuous, $\mathcal{A}(\cdot)$ can thus be regarded as a continuous function. In addition, $\mathcal{A}(\cdot)$ is a point-to-point mapping. Hence, $\mathcal{A}(\cdot)$ is closed at points outside $\boldsymbol{\Omega}$ (see Example 2 in Section 7.6 in Luenberger et al. (1984) ). The proof is completed. $\square$

## E    EQUIVALENT REALIZATION OF LSSMs

Based on the transformed coordinates, we can derive

$$\overline{\boldsymbol{x}}_t = \boldsymbol{\Phi}\boldsymbol{x}_t = \boldsymbol{\Phi}\boldsymbol{A}\boldsymbol{x}_{t-1} + \boldsymbol{\Phi}\boldsymbol{B}\boldsymbol{u}_t + \boldsymbol{\Phi}\boldsymbol{\varepsilon}_t = \left(\boldsymbol{\Phi}\boldsymbol{A}\boldsymbol{\Phi}^{-1}\right)\overline{\boldsymbol{x}}_{t-1} + \left(\boldsymbol{\Phi}\boldsymbol{B}\right)\boldsymbol{u}_t + \boldsymbol{\Phi}\boldsymbol{\varepsilon}_t, \quad (60)$$

$$\boldsymbol{y}_t = \boldsymbol{C}\boldsymbol{x}_t + \boldsymbol{D}\boldsymbol{u}_t + \boldsymbol{\omega}_t = \left(\boldsymbol{C}\boldsymbol{\Phi}^{-1}\right)\overline{\boldsymbol{x}}_t + \boldsymbol{D}\boldsymbol{u}_t + \boldsymbol{\omega}_t. \quad (61)$$

Hence, we can derive an equivalent realization of the original LSSM as follows:

$$\overline{\boldsymbol{x}}_t = \overline{\boldsymbol{A}}\overline{\boldsymbol{x}}_{t-1} + \overline{\boldsymbol{B}}\boldsymbol{u}_t + \overline{\boldsymbol{\varepsilon}}_t, \quad (62)$$

$$\boldsymbol{y}_t = \overline{\boldsymbol{C}}\overline{\boldsymbol{x}}_t + \boldsymbol{D}\boldsymbol{u}_t + \boldsymbol{\omega}_t, \quad (63)$$

where $\overline{\boldsymbol{A}} = \boldsymbol{\Phi}\boldsymbol{A}\boldsymbol{\Phi}^{-1}, \overline{\boldsymbol{B}} = \boldsymbol{\Phi}\boldsymbol{B}, \overline{\boldsymbol{C}} = \boldsymbol{C}\boldsymbol{\Phi}^{-1}$, and $\overline{\boldsymbol{\varepsilon}}_t = \boldsymbol{\Phi}\boldsymbol{\varepsilon}_t$. Because $\boldsymbol{\varepsilon}_t \sim \mathcal{N}\left(\boldsymbol{0}, \boldsymbol{R}\right)$, we can derive the mean of $\overline{\boldsymbol{\varepsilon}}_t$ as follows:

$$\mathbb{E}[\overline{\boldsymbol{\varepsilon}}_t] = \mathbb{E}[\boldsymbol{\Phi}\boldsymbol{\varepsilon}_t] = \boldsymbol{\Phi}\mathbb{E}[\boldsymbol{\varepsilon}_t] = \boldsymbol{0}. \quad (64)$$

In addition, the covariance of $\overline{\boldsymbol{\varepsilon}}_t$ can be derived as follows:

$$\mathbb{E}\left[\left(\overline{\boldsymbol{\varepsilon}}_t - \mathbb{E}[\overline{\boldsymbol{\varepsilon}}_t]\right)\left(\overline{\boldsymbol{\varepsilon}}_t - \mathbb{E}[\overline{\boldsymbol{\varepsilon}}_t]\right)'\right] = \mathbb{E}\left[\boldsymbol{\Phi}\boldsymbol{\varepsilon}_t\boldsymbol{\varepsilon}_t'\boldsymbol{\Phi}'\right] = \boldsymbol{\Phi}\mathbb{E}[\boldsymbol{\varepsilon}_t\boldsymbol{\varepsilon}_t']\boldsymbol{\Phi}' = \boldsymbol{\Phi}\boldsymbol{R}\boldsymbol{\Phi}'. \quad (65)$$

Hence, we have $\overline{\boldsymbol{\varepsilon}}_t \sim \mathcal{N}\left(\boldsymbol{0}, \boldsymbol{\Phi}\boldsymbol{R}\boldsymbol{\Phi}'\right)$.

## F    COMPUTATIONAL COMPLEXITY ANALYSIS

The high computational cost of the proposed algorithm primarily stems from the matrix inversion operations involved in closed-form updates. In each iteration, the expectation step entails the inversion of matrices of size $n \times n$ and $m \times m$ a total of $T$ times following Lemma 3.1. Hence, the computational complexity of the expectation step is mainly determined by $\mathcal{O}(Tn^3 + Tm^3)$. In the maximization step, we derive row-wise update rules for $\boldsymbol{A}, \boldsymbol{B}, \boldsymbol{C}$, and $\boldsymbol{D}$ to enable analytical update formulas. To update each row of $\boldsymbol{A}, \boldsymbol{B}, \boldsymbol{C}$, and $\boldsymbol{D}$, we need to calculate the inverses of the $n \times n$ and $p \times p$ matrices as shown in equation 20–equation 23. As a result, the computational complexity of the maximization step is dominated by $\mathcal{O}((m+n)(n^3 + p^3))$. Because the computational cost of the proposed algorithm scales at least cubically with respect to one of $m, n$, and $p$, it is hard to deal with the large-scale problems currently. Hence, our future work will explore how to mitigate the computational bottleneck to make it more applicable.

## G    THRESHOLD SELECTION

While the proposed algorithm is globally convergent, the generated sequence converges to a local maximum or saddle point of the posterior distribution only in the limit of infinite sequence length. Hence, many learned parameters in system matrices approach zero but never reach it exactly during algorithm implementation. To achieve accurate topology recovery, it is necessary to set the learned parameters below a predefined threshold to zero to avoid numerical errors. In particular, the learned parameters corresponding to the true zero components in the system matrices are typically several orders of magnitude smaller than those learned for the nonzero components. Hence, this pronounced difference in magnitude provides a wide and stable range for selecting the threshold.

Specifically, the threshold can be selected by visualizing how the number of nonzero components in learned system matrices, denoted as $N$, varies with the threshold. When the threshold is too small, all components of learned system matrices remain nonzero, $N$ is thus relatively large. As the threshold increases, the learned parameters corresponding to the true zero components are gradually set to zero, and $N$ decreases accordingly. However, due to the pronounced difference in magnitude between the learned parameters corresponding to the true zero components and those corresponding to the nonzero components, $N$ remains stable over a certain range of threshold values. Once the threshold exceeds this range, $N$ begins to decrease again because some learned parameters corresponding to true nonzero components are also set to zero. Therefore, the threshold can be safely selected from this stable interval to ensure reliable topology recovery. For example, Figure 2 illustrates the number of nonzero components in the learned system matrices under different threshold settings for the synthetic system in Section 5.1. The orange markers indicate the threshold values at which the proposed algorithm successfully learns the inherent topological structure among the variables, and the MRE remains unchanged. As such, we set the threshold to 0.005 in the experiment.

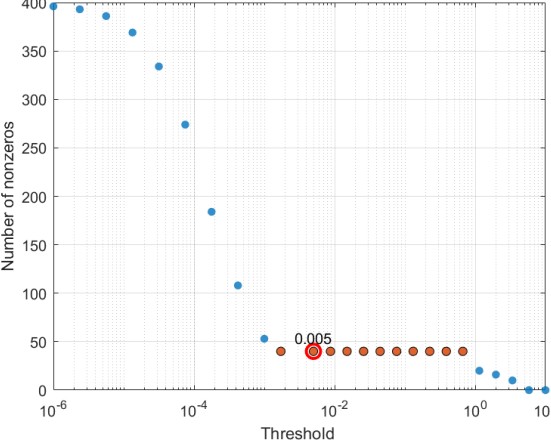

Figure 2: The number of nonzero components in the learned system matrices across various threshold settings. When the threshold lies within the interval marked by the orange dots, the proposed algorithm can successfully recover the inherent topological structure among the variables, and the MRE remains unchanged.

# H ADDITIONAL EXPERIMENTAL RESULTS ON THE SYNTHETIC SYSTEM

## H.1 EXPERIMENTAL RESULTS UNDER DIFFERENT INITIAL VALUES

To assess the sensitivity of the proposed algorithm to initial values, we further evaluate its performance on the synthetic system described in Section 5.1 by initializing the $A, B, C, D, R$, and $Q$ as $aI_{10}$, where $a$ varies from 0.6 to 1.4 in increments of 0.2.

Table 2 records whether the proposed algorithm successfully learns the topological structure of the system matrices and MRE defined in the paper. As observed from the table, the experimental results are consistent with those in Section 5.1 , indicating that the proposed algorithm is robust to initial values. Remarkably, even when the initial state transition matrix $A$ is unstable (i.e., $a > 1$), the proposed algorithm is still able to accurately learn the topological structure of the true system.

Table 2: Experimental results of the proposed algorithm on the 10-dimensional synthetic system across different initial values

| $a$ | 0.6 | 0.8 | 1 | 1.2 | 1.4 |
|---|---|---|---|---|---|
| Success? | $\checkmark$ | $\checkmark$ | $\checkmark$ | $\checkmark$ | $\checkmark$ |
| MRE | 5.70% | 5.70% | 5.70% | 5.70% | 5.69% |

## H.2 EXPERIMENTAL RESULTS OF INDEPENDENT TRIALS

Here, we conduct 20 independent trials to demonstrate that the proposed algorithm is stable across multiple runs. Specifically, we set the random seed to increase evenly from 1 to 20. Table 3 reports the success rates of all the algorithms in learning the topological structure among the variables and the average MRE. Compared to the classical algorithms, only the proposed algorithm successfully learns the inherent topological structure among the variables in almost all cases. In addition, the proposed algorithm can achieve a 100% success rate by slightly increasing the threshold below which learned parameters are set to zero.

Table 3: Experimental results of all the algorithms on 20 independent trails

| Method | Ours | LSM_PEM | LSM_HK | 4SID | MLE |
|---|---|---|---|---|---|
| Success rate | **95%** | 0 | 0 | 0 | 0 |
| Average MRE | **5.67%** | 19.63% | 21.56% | 20.37% | 5.97% |

# I EXPERIMENTAL RESULTS ON THE NON-DIAGONAL AND NON-INVERTIBLE SYNTHETIC SYSTEMS

## I.1 NON-DIAGONAL SYSTEM

Here, we further test the proposed algorithm on a non-diagonal system to illustrate its effectiveness. To generate non-diagonal system matrices, we randomly set one component per row to 0.8 in $A$, and to 2 in $B$, $C$, and $D$, with all other elements set to zero. Particularly, the nonzero elements are deliberately placed to ensure that $A, B, C$, and $D$ maintain full rank. All other experimental settings remain the same as in Section 5.1.

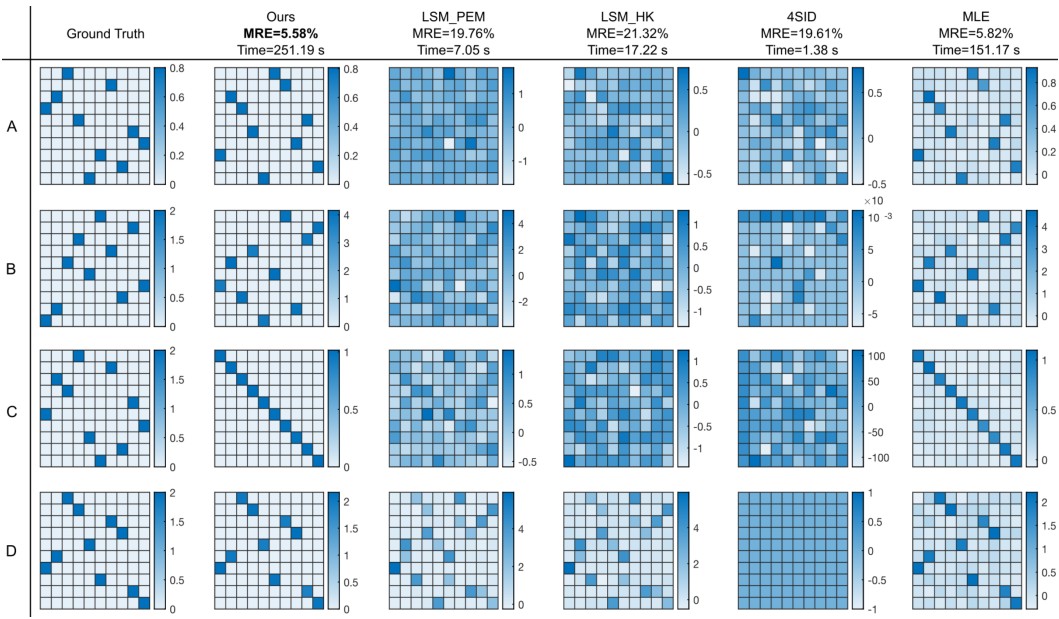

Figure 3: Experimental results of all the algorithms on the non-diagonal systems.

Figure 3 records the experimental results of all the algorithms on the non-diagonal system. Due to the similarity transformation as discussed in Section 4, the learned $A$, $B$, and $C$ of all the algorithms differ in form from the ground truth. Unlike classical algorithms, however, the proposed algorithm restricts the nonsingular matrix of the similarity transformation to be a generalized permutation matrix. By comparing the learned $B$ and $C$ with ground truth, we can derive the nonsingular matrix as follows:

$$\Phi \approx \begin{bmatrix} 0 & 0 & 0 & 2 & 0 & 0 & 0 & 0 & 0 & 0 \\ 0 & 0 & 0 & 0 & 0 & 0 & 2 & 0 & 0 & 0 \\ 0 & 2 & 0 & 0 & 0 & 0 & 0 & 0 & 0 & 0 \\ 0 & 0 & 2 & 0 & 0 & 0 & 0 & 0 & 0 & 0 \\ 0 & 0 & 0 & 0 & 0 & 0 & 0 & 0 & 2 & 0 \\ 2 & 0 & 0 & 0 & 0 & 0 & 0 & 0 & 0 & 0 \\ 0 & 0 & 0 & 0 & 0 & 0 & 0 & 0 & 0 & 2 \\ 0 & 0 & 0 & 0 & 2 & 0 & 0 & 0 & 0 & 0 \\ 0 & 0 & 0 & 0 & 0 & 0 & 0 & 2 & 0 & 0 \\ 0 & 0 & 0 & 0 & 0 & 2 & 0 & 0 & 0 & 0 \end{bmatrix}. \tag{66}$$

This implies that the discrepancy between the learned system matrices of the proposed algorithm and ground truth stems from a different ordering and scaling of system states. Hence, it is easy to check that the proposed algorithm still accurately captures the underlying system dynamics and preserves the inherent topological structure among the variables.

## I.2 NON-INVERTIBLE SYSTEM

To evaluate the proposed algorithm on the non-invertible system, we replace the system matrix $\boldsymbol{A}$ in the synthetic system of Section 5.1 with a singular matrix. Specifically, we first construct an anti-diagonal matrix with all nonzero components set to 0.6, and then replace its seventh and eighth rows with the following vector: [0 0 0.6 0.6 0 0 0 0 0 0]. As such, the seventh and eighth rows of $\boldsymbol{A}$ are identical, making the matrix non-invertible. Besides, all other experimental settings remain the same as in Section 5.1. Figure 4 reports the experimental results of all the algorithms on the non-invertible system. Similarly, we can derive $\boldsymbol{\Phi} \approx 2\boldsymbol{I}_{10}$ by comparing the learned system matrices $\boldsymbol{B}$ and $\boldsymbol{C}$ with the ground truth. Hence, the proposed algorithm still successfully preserves the topological structure of system matrices, and obtains the lowest MRE compared to the other algorithms.

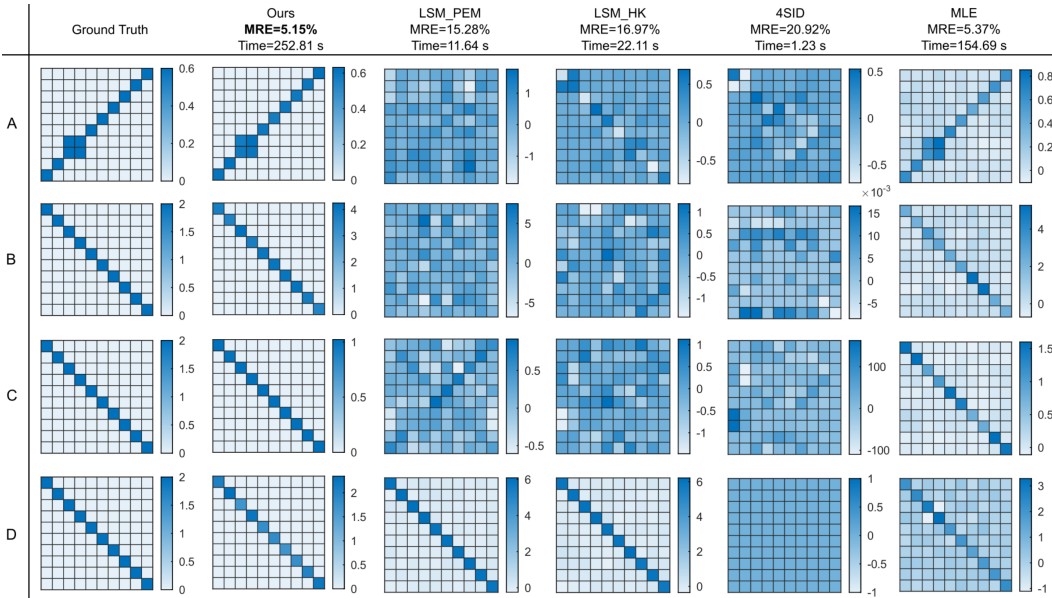

Figure 4: Experimental results of all the algorithms on the non-invertible system.

## J    THE USE OF LARGE LANGUAGE MODELS (LLMS)

In accordance with the ICLR policy on the use of large language models (LLMs), we declare that LLMs are only employed as a writing assistant to polish the language of this paper. They do not contribute to the research ideation, experimental design, data analysis, or interpretation of the results.

