# OpenReview forum: "Learning linear state-space models with sparse system matrices"
_ICLR.cc/2026/Conference — ICLR 2026 Poster_

### Official Review · Reviewer_jpHJ · 2025-10-23

**Soundness:** 3
**Presentation:** 3
**Contribution:** 3
**Rating:** 6
**Confidence:** 4

**Summary:**

This paper addresses the problem of learning linear state-space models (LSSMs) with sparse system matrices, a setting relevant to many real-world dynamical systems where interactions are limited. The authors propose a maximum a posteriori (MAP)–EM algorithm that introduces Student-t sparsity-promoting priors on the system matrices and derives closed-form row-wise updates for all parameters. The algorithm is theoretically supported by a convergence guarantee in the EM sense (Theorem 3.3) and empirically evaluated on both synthetic and industrial datasets.

**Strengths:**

1. **Well-motivated problem.**
Learning LSSMs with sparse system matrices is both practically and theoretically relevant, as sparsity naturally arises in many real systems with limited or localized interactions.

2. **Clear algorithm design and convergence guarantee.**
The proposed MAP–EM algorithm is clearly described, with detailed derivations and an explicit convergence guarantee (Theorem 3.3), which adds credibility to the approach.

**Weaknesses:**

1. **Lack of a formal sparsity definition.**
 “Sparse system matrices” are introduced implicitly through element-wise priors rather than through a clear structural definition or quantitative metric for topology recovery. As a result, the structural claims rely mainly on visual inspection rather than measurable topology-level metrics.

2. **Limited theoretical support for the “generalized permutation” claim.** The conclusion that the admissible similarity transformation $\Phi$ reduces to a generalized permutation matrix is demonstrated only through an example, without a general proof or stated conditions under which this holds.

3. **Potential unfairness in sparsity comparisons.** The authors apply post-hoc thresholding (“parameters below a predefined threshold are truncated to zero”) to their own results to emphasize sparsity, but it is unclear whether the same was done for the MLE baseline. Given that in Figures 1 and 2 the MLE method already yields a structure and MRE very close to the proposed approach, applying the same thresholding to MLE might lead to similarly sparse and accurate results, thus weakening the claimed structural advantage.

**Questions:**

1. **Thresholding fairness:** If the same post-hoc thresholding used in your method is applied to the MLE estimates, how do the resulting structure plots and MRE values change? Please report results for “MLE + threshold” using the identical threshold schedule.

2. **Threshold sensitivity:** How sensitive are the topology recovery and MRE results to the chosen truncation threshold? How is this threshold selected in practice?

3. **Non-invertible systems:** The examples presented involve invertible $A$ matrices. Can your proposed algorithm be applied to systems where $A$ is singular (non-invertible)? If yes, please provide an experiment to illustrate this case.

4. **Hyperparameter robustness:** How sensitive are your results to the Student-t hyperparameters ($a_0, b_0$) and to the initialization of $\Gamma$? Do you have recommended defaults or practical guidance for choosing these values?

---

> ### Author Response · Authors · 2025-11-20
> **Rebuttal #1**
>
> We thank the reviewer for taking the time to review our paper. We are glad that you think the paper is well-motivated and the proposed algorithm is clearly presented with an explicit convergence guarantee.
>
> **Response to Weaknesses**
> >**Lack of a formal sparsity definition. “Sparse system matrices” are introduced implicitly through element-wise priors rather than through a clear structural definition or quantitative metric for topology recovery. As a result, the structural claims rely mainly on visual inspection rather than measurable topology-level metrics.**
>
> - We agree with the reviewer that a structural definition of sparse system matrices is necessary for topology recovery.  To this end, we consider the following two perspectives to provide such a formal definition. First, the linear state-space models (LSSMs) with sparse system matrices  should include the minimally required parameters to explain time-series data following the Occam’s razor principle, which favors explanations constructed with the smallest possible set of elements. Second, due to the similarity transformation, LSSMs  admit many equivalent representations with different levels of sparsity.  However, the proposed algorithm learns system matrices by balancing model complexity and modeling error. Hence, it also focuses on learning the sparse system matrices with  minimally required parameters.  Because the $\ell_0$ norm counts the number of nonzero components in a matrix, we have given the following definition for the LSSMs with sparse system matrices to ensure identifiability in Section 3 of the revised version.
>
>   _Due to the similarity transformation, LSSMs admit many equivalent representations with different levels of sparsity, where the corresponding transformed system matrices are given by  $ \Phi A \Phi^{-1}$, $ \Phi B$, $ C \Phi^{-1}$, and $ D$, with $ \Phi\in\mathbb R^{n\times n}$ being a nonsingular matrix. However, we focus on learning the LSSMs with sparse system matrices that include minimally required parameters in accordance with the Occam’s razor principle. Hence, we define the identifiability of LSSMs to ensure that the resulting ambiguities can only be permutations and scaling, as is formalized as follows._
>
>   _Definition 3.1: (Identifiability) For LSSMs with nonzero system matrices $ A,  B,  C$, and $ D$,  if any nonsingular matrix $ \Phi\in \mathbb R^{n\times n}$ satisfying_
>
>   $\||\Phi A\Phi^{-1}\||_0= \|| A\||_0, \||\Phi B\||_0= \|| B\||_0,$ and $\||C \Phi^{-1}\||_0=\||C\||_0$
>
>   _must be a generalized permutation matrix, then such systems are said to be essentially identifiable, up to permutation and scaling._
>
> >**Limited theoretical support for the “generalized permutation” claim. The conclusion that the admissible similarity transformation $ \Phi$ reduces to a generalized permutation matrix is demonstrated only through an example, without a general proof or stated conditions under which this holds.**
>
> - To make the claim more rigorous, we have formalized the definition of identifiability for LSSMs with sparse system matrices in the revised version.  Under this definition, if the system is identifiable, the nonsingular matrix $ \Phi$ must be a  generalized permutation matrix to ensure that the number of nonzero components in the system matrices remains unchanged.
>
> >**Potential unfairness in sparsity comparisons. The authors apply post-hoc thresholding (“parameters below a predefined threshold are truncated to zero”) to their own results to emphasize sparsity, but it is unclear whether the same was done for the MLE baseline. Given that in Figures 1 and 2 the MLE method already yields a structure and MRE very close to the proposed approach, applying the same thresholding to MLE might lead to similarly sparse and accurate results, thus weakening the claimed structural advantage.**
>
> - We appreciate the reviewer’s comment regarding thresholding fairness. We feel sorry for causing this misunderstanding due to the lack of a detailed description of the experimental setup. In fact, the same post-hoc thresholding procedure is applied to all algorithms to ensure a fair comparison. Hence, the experimental results in the paper have reported results for  the other  algorithms using an identical threshold schedule. In the revised version, we have added the following sentence in Experiment to avoid any misunderstanding : _To ensure a fair comparison, the learned parameters of all the other algorithms that fall below this threshold are likewise set to zero._

---

> ### Author Response · Authors · 2025-11-20
> **Rebuttal #2**
>
> **Response to Questions**
>
> >**Thresholding fairness: If the same post-hoc thresholding used in your method is applied to the MLE estimates, how do the resulting structure plots and MRE values change? Please report results for “MLE + threshold” using the identical threshold schedule.**
>
> - See the response to Weaknesses.
>
> >**Threshold sensitivity: How sensitive are the topology recovery and MRE results to the chosen truncation threshold? How is this threshold selected in practice?**
>
> - **Algorithm’s robustness to the threshold.**  While the proposed algorithm is globally convergent, the generated sequence converges to a local maximum or saddle point of the posterior distribution only in the limit of infinite sequence length.  Hence, many learned parameters in system matrices approach zero but never reach it exactly during algorithm implementation. To achieve accurate topology recovery, it is necessary to set the learned parameters below a predefined threshold to zero to avoid numerical errors. In particular, the learned parameters corresponding to the true zero components  in the system matrices are typically several orders of magnitude smaller than those learned for the nonzero components. Hence, this pronounced difference in magnitude provides a wide and stable range for selecting the threshold, making the topology recovery and MRE insensitive to its exact value. For example,  when the threshold is selected within the approximate range of $10^{-3}$ to  $10^{-1}$ in the experiment of Section 5.1, the proposed algorithm can successfully learn the inherent topological structure
> among the variables, and the MRE remains unchanged.
>
> - **Threshold selection.** In practice, the threshold can be selected by visualizing how the number of nonzero components in learned system matrices, denoted as $N$, varies with the threshold. When the threshold is too small, all components of  learned system matrices remain nonzero,  $N$ is thus relatively large. As the threshold increases, the learned parameters corresponding to the true zero components are gradually set to zero, and $N$ decreases accordingly. However, due to the pronounced difference in magnitude between the learned parameters corresponding to the true zero components and those corresponding to the nonzero components,
> $N$ remains stable over a certain range of threshold values. Once the threshold exceeds this range,  $N$ begins to decrease again because some learned parameters corresponding to true nonzero components are also set to zero. Therefore, the threshold can be safely selected from this stable interval to ensure reliable topology recovery.
>
> - Based on your valuable comment, we have added a discussion on threshold selection and demonstrated the algorithm’s robustness to the threshold in Appendix G in the revised version.

---

> ### Author Response · Authors · 2025-11-20
> **Rebuttal #3**
>
> >**Non-invertible systems: The examples presented involve invertible  matrices. Can your proposed algorithm be applied to systems where  $A$ is singular (non-invertible)? If yes, please provide an experiment to illustrate this case.**
>
> - Because the proposed algorithm does not rely on any operation that requires $ A$ to be invertible, it can also be applied to systems where $ A$ is singular. To address the reviewer’s concern, we further conduct an additional experiment in which only the matrix $ A$ in the synthetic system of Section 5.1 is replaced with a singular matrix. Specifically, we first construct an anti-diagonal matrix with all nonzero entries set to 0.6, and then replace its  seventh and eighth rows with the following vector:
> [0 0 0.6 0.6 0 0 0 0 0 0].
>  As such,  the seventh and eighth rows of $ A$ are identical, making the matrix singular. The following table records whether the algorithm successfully learns the topological structure of the system matrices and MRE defined in the paper.
>   |Method | Ours  | LSM\_PEM  | LSM\_ HK | 4SID | MLE |
>   |-----|:------:|:---:|:---:|:------:|:------:|
>   | Success   | $\checkmark$  | $\times$  |$\times$ |  $\times$  |  $\times$ |
>   | MRE   | **5.15\%**   | 15.28\%   |16.97\% | 20.90\%  |5.37\% |
>
>   Hence, experimental results on the non-invertible matrix $A$ are consistent with those presented in the paper. The proposed algorithm preserves the inherent topological structure among the variables and achieve the lowest MRE compared to the other algorithms.
>
> - Based on your valuable comment, we have included an additional experiment on the non-invertible case to further demonstrate the effectiveness of the proposed algorithm in Appendix I.2 of the revised version.
>
> >**Hyperparameter robustness: How sensitive are your results to the Student-t hyperparameters ($a_0,b_0$) and to the initialization of $\Gamma$? Do you have recommended defaults or practical guidance for choosing these values?**
>
>
> - Basically, we hope the priors/hyperpriors  to be flat to ensure that all possible values have roughly equal probability initially. Hence, $a_0$ and $b_0$ in the  Inverse-Gamma distribution are typically set to very small  values (e.g., $10^{-5}-10^{-7}$) to generate non-informative hyperprior as discussed in Section 3.1.   For each component of system matrices, we impose a Gaussian prior on it with mean zero and variance governed by the corresponding component of $\Gamma$. To ensure the Gaussian prior remains as flat as possible, each component of $ \Gamma$ should be chosen to be moderately large  (e.g., $10^4-10^6$).
>
>  - To further address your concern, we conduct two additional sets of experiments on the synthetic system in Section 5.1  under different experimental settings to  validate the algorithm’s robustness. First, all other experimental settings  remain  the same as in Section 5.1, while $a_0$ and $b_0$ are varied across three values: $10^{-5}, 10^{-6}$, and $10^{-7}$. The following table reports the experimental results under different hyperparameters:
>
>     |$a_0=b_0$ | $10^{-5}$  | $10^{-6}$  | $10^{-7}$ |
>     |-----|:------:|:---:|:---:|
>     | Success   | $\checkmark$  | $\checkmark$ |$\checkmark$ |
>     | MRE   | 5.70\%  | 5.60\%   |5.60\% |
>
>
>     In contrast, the second set of  experiments varies each component of $\Gamma$ across three values:$10^{4}, 10^{5}$, and $10^{6}$, with experimental results reported in the following table:
>
>     |$\Gamma$ | $10^{4}$  | $10^{5}$  | $10^{6}$ |
>     |-----|:------:|:---:|:---:|
>     | Success   | $\checkmark$  | $\checkmark$ |$\checkmark$ |
>     | MRE   | 5.70\%  | 5.70\%   |5.70\% |
>
>     Overall, the proposed algorithm consistently learns the topological structure of the system matrices, and the MRE remains stable across all experimental settings. Hence,  the experimental results demonstrate that the proposed  algorithm is robust to the choice of $a_0$, $b_0$, and initialization of  $\Gamma$.

---

> > ### Comment · Reviewer_jpHJ · 2025-11-24
> >
> > I thank the authors for their rebuttal. They have addressed all of my concerns, and I will raise my score.

---

> > > ### Author Response · Authors · 2025-11-24
> > > **Response to Reviewer jpHJ**
> > >
> > > We appreciate the reviewer for the thoughtful feedback and for raising the score. We are glad to know that our responses resolved your concerns.

---

### Official Review · Reviewer_ggGN · 2025-10-25

**Soundness:** 3
**Presentation:** 3
**Contribution:** 3
**Rating:** 6
**Confidence:** 3

**Summary:**

This paper introduces a method to learn sparse linear state-space models. This is achieved by a hierarchical Student's t-distribution prior on system matrices which results in sparsity bias. The system matrices, hyperprior parameters and noises are then iteratively optimized via an expectation-maximization procedure. For each minimization step, the authors derive closed-form update rules based on block-coordinate descent on each parameter. They validate their results on some synthetic and real-world systems.

**Strengths:**

The paper seems to be a natural extension of prior work on sparse bayesian learning (e.g., of the ideas in Wang et al., 2024). I am not sure how original the technical ideas are but the paper is well-executed and derives the necessary results in a more complicated setting.

The technical results are complete in the sense that there seems to be nothing left on the table given the introduced setting. Additional improvements can only be made by imposing other constraints or relaxed versions of the algorithm for computational efficiencies which I find out of the scope. Therefore, I consider the theoretical results complete and good quality. The ideas and the results are presented very clearly.

**Weaknesses:**

1. The results are promising empirically but more can be done if the authors think the algorithm presented is practical. For example, they can include runtime comparisons.
2. I think the paper puts the right results together from previous literature and has rather minimal technical contributions.

**Questions:**

1. Can the authors discuss technical novelties (if there is any) compared to the related work?
2. Can you discuss the computational requirements a bit more in depth compared to other algorithms? Isn't it normal that each iteration takes $T$ time as that's the size of the input. Why is it more computationally expensive? Is it the inverse operations in the closed-form updates?

---

> ### Author Response · Authors · 2025-11-20
> **Rebuttal #1**
>
> We thank the reviewer for taking the time to review our paper. We are glad that you found the technical results to be complete and of good quality.
>
> **Response to Weaknesses**
> >**The results are promising empirically but more can be done if the authors think the algorithm presented is practical. For example, they can include runtime comparisons.**
>
> - Based on your constructive comment, we have included  a detailed runtime comparison between the proposed algorithm and several standard alternatives across all experiments in the revised version. For your reference, the  runtimes (in seconds) of each experiment are summarized in the table below:
>
>   | Runtime (s)  | Ours  | LSM\_PEM  | LSM\_ HK | 4SID | MLE |
>   |-----|:------:|:---:|:---:|:------:|:------:|
>   | Diagonal system (Section 5.1)  | 249.14   | 9.35  |18.45 |  3.92  |  152.34 |
>   | Evaporation system  (Section 5.2) | 230.39   | 0.98   |13.23 | 0.50  |168.83 |
>   | Glass furnace (Section 5.2)  | 45.62   | 0.52   |6.20 | 0.29  |33.36 |
>   | Steam generator  (Section 5.2) | 441.88   | 4.94   |84.52 | 0.58  |299.10 |
>   | Non-diagonal system  (Appendix I.1)  | 251.19   | 7.05   |17.22 | 1.38  | 151.17 |
>   | Non-invertible system  (Appendix I.2)   | 252.81   | 11.64   |22.12 | 1.23  | 154.69 |
>
>   Compared with other algorithms, the proposed algorithm involves a large number of matrix inversions in each iteration. Hence, its runtime is  higher than that of the other algorithms. Nevertheless, this is also reasonable because we cannot expect an algorithm to achieve  both computational efficiency and superior learning performance in general. As discussed in Section 6, we will explore how to reduce computational complexity of the proposed algorithm in future work to make it more applicable.
>
> >**I think the paper puts the right results together from previous literature and has rather minimal technical contributions.**
>
> - We understand your concerns regarding the novelty of the work, and would like to take this opportunity to further clarify the technical novelties presented in our study.
>
>    **Algorithmic novelty.** As discussed in Section 2, while sparse Bayesian learning (SBL) has been applied to learn various dynamical systems, existing studies primarily focus on scenarios where  system states are directly measurable. For linear state-space models (LSSMs), however, system states   are unobservable, making it infeasible to naturally extend SBL to learn system matrices. In fact, learning LSSMs with sparse system matrices involves two fundamental problems: **state estimation** and **system identification**.  By exploring the expectation-maximization algorithm, this paper proposes a unified framework that seamlessly integrates the Rauch-Tung-Striebel (RTS) smoother for estimating hidden system states in the expectation step and SBL  for learning sparse system matrices in the maximization step. Consequently, the proposed algorithm enables learning LSSMs with sparse system matrices without prior knowledge of the system states. In addition, this paper designs a row-wise update strategy for system matrices to derive closed-form solutions, making the algorithm more straightforward to implement and analyze.
>
>   **Theoretical novelty.** Leveraging the Global Convergence Theorem, this paper demonstrates that the proposed algorithm for learning LSSMs with sparse system matrices is globally convergent,  indicating that the generated sequence converges to a local maximum or saddle point of the joint posterior distribution.

---

> ### Author Response · Authors · 2025-11-20
> **Rebuttal #2**
>
> **Response to Questions**
>
> >**Can the authors discuss technical novelties (if there is any) compared to the related work?**
>
> - See the response to Weaknesses.
>
> >**Can you discuss the computational requirements a bit more in depth compared to other algorithms? Isn't it normal that each iteration takes $T$ time as that's the size of the input. Why is it more computationally expensive? Is it the inverse operations in the closed-form updates?**
>
> - Based on your valuable comment, we have  provided a computational complexity analysis of the proposed algorithm in Appendix F of the revised version to illustrate why the proposed algorithm is computationally expensive as follows:
>
>   _The high computational cost of the proposed algorithm primarily stems from the matrix inversion operations involved in closed-form updates. In each iteration, the expectation step entails the inversion of matrices of size $n\times n$ and $m\times m$ a total of $T$ times following Lemma 3.1, where $n$ is the dimension of the hidden state, $m$ is the dimension of the output data, and $T$ is the length of the input data. Hence, the computational complexity of the expectation step is mainly determined by $\mathcal{O}(Tn^3+Tm^3)$. In the maximization step, we derive  row-wise update rules for $ A,  B,  C,$ and $ D$ to enable  analytical update formulas. To update each row of $ A,  B,  C,$ and $ D$, we need to calculate the inverses of the $n\times n$ and $p\times p$ matrices as shown in equation 20-equation 23, where $p$ is the dimension of the input data. As a result, the computational complexity of the maximization step is dominated by  $\mathcal{O}((m+n)(n^3+p^3))$. Because the computational cost of the proposed algorithm scales at least cubically with respect to one of  $m, n,$ and $p$, it is hard to deal with the large-scale problems currently. Hence, our future work will explore how to mitigate the computational bottleneck to make it more applicable._

---

> > ### Comment · Reviewer_ggGN · 2025-11-26
> >
> > I thank the authors for their rebuttal. I maintain my initial positive score.

---

> > > ### Author Response · Authors · 2025-11-26
> > > **Response to Reviewer ggGN**
> > >
> > > We deeply appreciate your prompt feedback and positive evaluation of our paper!

---

### Official Review · Reviewer_ttg7 · 2025-10-31

**Soundness:** 3
**Presentation:** 3
**Contribution:** 2
**Rating:** 6
**Confidence:** 3

**Summary:**

This paper proposes a new EM algorithm for learning linear state-space models (LSSMs) with explicitly sparse system matrices, a critical aspect for interpretability and efficiency in various scientific domains. The authors impose sparsity-promoting Student’s t-distribution priors on system matrices and develop a principled MAP estimation framework. The resulting EM procedure leverages the Rauch-Tung-Striebel smoother in the E-step and block coordinate descent in the M-step, with global convergence to local MAP optima proven via the Global Convergence Theorem. The paper claims its approach avoids the notorious similarity transformation pitfalls of classical LSSM learning algorithms, thus better recovering the true sparse topology. Experiments on synthetic and several real-world benchmark datasets show improvements in prediction accuracy and topological interpretability compared to established baselines.

**Strengths:**

The paper demonstrates methodological rigor through a principled MAP-EM formulation for LSSMs under sparsity priors, with complete derivations, update steps, and connections to established statistical techniques. Its theoretical soundness is reinforced by a formally proven convergence guarantee via the Global Convergence Theorem. The work explicitly addresses the similarity transformation issue, ensuring identifiability and interpretability by restricting transformations to generalized permutation matrices. The experiments on synthetic and real-world industrial datasets validate the method’s effectiveness, with figures and tables clearly illustrating preservation of topological structures and improved quantitative performance. The paper’s mathematical transparency and detailed derivations enhance reproducibility and clarity.

**Weaknesses:**

- The algorithm's reliance on full EM with smoothing and coordinate updates (Algorithm 1) and complexity discussed in Section 6, makes it computationally intensive. The paper only briefly mentions the limitation with respect to large-scale settings, without offering profiling, timing, or strategies for acceleration (e.g., stochastic EM, parallelization).

- As stated in Section 5.1 and experimental appendices, sparsity is only enforced "up to a predefined threshold" with truncation to accelerate convergence. While this is common in Bayesian learning, an explicit analysis of how the choice of this threshold affects success rates, stability, and meaningful recovery of structures is not present, nor is there a discussion of methods that enforce exact zeros by construction.

- The argument that sparsity-promoting priors restrict similarity transformations to generalized permutation matrices is supported only for very specific classes of topologies; most of the argument is illustrated rather than proven generally. Real-world systems with denser interconnections may still pose identifiability challenges; this should be discussed further.

- While the paper is mathematically rigorous, much of the machinery (Student’s $t$ prior for sparsity, hierarchical Bayesian treatment, EM updates, block coordinate descent, even the use of convergence theorems) is largely a synthesis/extension of established sparse Bayesian learning and system identification literature. The main differentiating insight (identifiability under sparsity priors limiting transformations to permutation/scaling) though interesting, is not deeply explored beyond special-case analysis.

- Although synthetic and 3 real-world datasets are experimented on, comparison to more recent or advanced sparse LSSM learning methods, such as those using Lasso or regularized MLE approaches (e.g., [1]), is missing. Moreover, evaluation is limited mostly to MRE; the case for learning the "correct" topology is largely qualitative except for synthetic scenarios, which could be expanded quantitatively (e.g., with precision/recall on support recovery).

**Refs:**

[1] Fattahi, Salar, Nikolai Matni, and Somayeh Sojoudi. "Learning sparse dynamical systems from a single sample trajectory." 2019 IEEE 58th Conference on Decision and Control (CDC). IEEE, 2019.

**Questions:**

I would wish author(s) can clearify my concerns around thresholding for sparsity, hyperparameter selection, and comparison to Lasso/regularized MLE-style estimators.

The constraints on similarity transformations in the presence of sparsity-promoting priors are presented for special cases. Can these insights be generalized, or are there pathological systems where the identifiability argument fails?

---

> ### Author Response · Authors · 2025-11-20
> **Rebuttal #1**
>
> We thank the reviewer for carefully reviewing our manuscript and acknowledging our contributions.
>
> **Response to Weaknesses**
> > **The algorithm's reliance on full EM with smoothing and coordinate updates (Algorithm 1) and complexity discussed in Section 6, makes it computationally intensive. The paper only briefly mentions the limitation with respect to large-scale settings, without offering profiling, timing, or strategies for acceleration (e.g., stochastic EM, parallelization).**
>
> - Based on your valuable comment, we have provided a computational complexity analysis of the proposed algorithm in Appendix F and included a detailed runtime comparison with several standard alternatives across all experiments in the revised version.
>
>   **Computational complexity analysis**. The high computational cost of the proposed algorithm primarily stems from the matrix inversion operations involved in closed-form updates. In each iteration, the expectation step entails the inversion of matrices of size $n\times n$ and $m\times m$ a total of $T$ times following Lemma 3.1, where $n$ is the dimension of the hidden state, $m$ is the dimension of the output data, and $T$ is the length of the input data. Hence, the computational complexity of the expectation step is mainly determined by $\mathcal{O}(Tn^3+Tm^3)$. In the maximization step, we derive  row-wise update rules for $ A,  B,  C,$ and $ D$ to enable  analytical update formulas. To update each row of $ A,  B,  C,$ and $ D$, we need to calculate the inverses of the $n\times n$ and $p\times p$ matrices as shown in equation 20-equation 23, where $p$ is the dimension of the input data. As a result, the computational complexity of the maximization step is dominated by  $\mathcal{O}((m+n)(n^3+p^3))$. Because the computational cost of the proposed algorithm scales at least cubically with respect to one of  $m, n,$ and $p$, it is hard to deal with the large-scale problems currently. Hence, our future work will explore how to mitigate the computational bottleneck to make it more applicable.
>
>   **Runtime comparison.** For your reference, the  runtimes (in seconds) of each experiment are summarized in the table below:
>
>   | Runtime (s)  | Ours  | LSM\_PEM  | LSM\_ HK | 4SID | MLE |
>   |-----|:------:|:---:|:---:|:------:|:------:|
>   | Diagonal system (Section 5.1)  | 249.14   | 9.35  |18.45 |  3.92  |  152.34 |
>   | Evaporation system  (Section 5.2) | 230.39   | 0.98   |13.23 | 0.50  |168.83 |
>   | Glass furnace (Section 5.2)  | 45.62   | 0.52   |6.20 | 0.29  |33.36 |
>   | Steam generator  (Section 5.2) | 441.88   | 4.94   |84.52 | 0.58  |299.10 |
>   | Non-diagonal system  (Appendix I.1)  | 251.19   | 7.05   |17.22 | 1.38  | 151.17 |
>   | Non-invertible system  (Appendix I.2)   | 252.81   | 11.64   |22.12 | 1.23  | 154.69 |
>
>   Compared with the other algorithms, the proposed algorithm involves a large number of matrix inversions in each iteration. Hence, its runtime is   higher than that of the other algorithms. Nevertheless, this is also reasonable because we cannot expect an algorithm to achieve  both computational efficiency and superior learning performance in general.
>
>   **Acceleration strategy.** In the revised version, we have added a sentence in Discussion to indicate that the stochastic EM algorithm can be employed to reduce  computation time as follows: _Hence, our future work will focus on reducing computation time to make the proposed algorithm applicable to large-scale settings. For instance, we can adopt the stochastic EM algorithm to  reduce the computational cost by using a mini-batch of  data instead of the full batch  during the expectation step._

---

> ### Author Response · Authors · 2025-11-20
> **Rebuttal #2**
>
> > **As stated in Section 5.1 and experimental appendices, sparsity is only enforced ”up to a predefined threshold” with truncation to accelerate convergence. While this is common in Bayesian learning, an explicit analysis of how the choice of this threshold affects success rates, stability, and meaningful recovery of structures is not present, nor is there a discussion of methods that enforce exact zeros by construction.**
>
> - Based on your valuable comment, we have added a discussion on threshold selection and demonstrated the algorithm’s robustness to the threshold in Appendix G of the revised version.
>
>   **Algorithm’s robustness to the threshold.** While the proposed algorithm is globally convergent,  the generated sequence  converges to a local maximum or saddle point of the posterior distribution only in the limit of infinite sequence length. Hence, many learned parameters in system matrices approach zero but never reach it exactly during algorithm implementation. To achieve accurate topology recovery, it is necessary to set the learned parameters below a predefined threshold to zero to avoid numerical errors. In particular, the learned parameters corresponding to the true zero components  in  system matrices are typically several orders of magnitude smaller than those learned for the nonzero components. Hence, this pronounced difference in magnitude provides a wide and stable range for selecting the threshold, making the topology recovery and MRE insensitive to its exact value. For example,  when the threshold is selected within the approximate range of $10^{-3}$ to  $10^{-1}$ in the experiment of Section 5.1, the proposed algorithm can successfully learn the inherent topological structure among the variables, and the MRE remains unchanged.
>
>
>
>     **Threshold selection.** In practice, the threshold can be selected by visualizing how the number of nonzero components in system matrices, denoted as $N$, varies with the threshold. When the threshold is too small, all components of system matrices remain nonzero,  $N$ is thus relatively large. As the threshold increases, the learned parameters corresponding to the true zero components in system matrices are gradually set to zero, and $N$ decreases accordingly. However, due to the pronounced difference in magnitude between the learned parameters corresponding to the true zero components and those corresponding to the nonzero components,  $N$ remains stable over a certain range of threshold values. Once the threshold exceeds this range,  $N$ begins to decrease again because some learned parameters corresponding to true nonzero components are also set to zero. Therefore, the threshold can be safely selected from this stable interval to ensure reliable topology recovery.
>
>
> >**The argument that sparsity-promoting priors restrict similarity transformations to generalized permutation matrices is supported only for very specific classes of topologies; most of the argument is illustrated rather than proven generally. Real-world systems with denser interconnections may still pose identifiability challenges; this should be discussed further.**
>
>
> - We agree with the reviewer that the LSSMs with denser interconnections may pose identifiability challenges.  Hence, it is necessary to impose a structural assumption on LSSMs with sparse system matrices to ensure identifiability. In the revised version, we have formalized the definition of identifiability for LSSMs with sparse system matrices in Section 3 as follows:
>
>   _Due to the similarity transformation, LSSMs admit many equivalent representations with different levels of sparsity, where the corresponding transformed system matrices are given by  $ \Phi A \Phi^{-1}$, $ \Phi B$, $ C \Phi^{-1}$, and $ D$, with $ \Phi\in\mathbb R^{n\times n}$ being a nonsingular matrix. However, we focus on learning the LSSMs with sparse system matrices that include minimally required parameters in accordance with the Occam’s razor principle. Hence, we define the identifiability of LSSMs to ensure that the resulting ambiguities can only be permutations and scaling, as is formalized as follows._
>
>   _Definition 3.1: (Identifiability) For LSSMs with nonzero system matrices $ A,  B,  C$, and $ D$,  if any nonsingular matrix $ \Phi\in \mathbb R^{n\times n}$ satisfying_
>
>   $\||\Phi A\Phi^{-1}\||_0= \|| A\||_0, \||\Phi B\||_0= \|| B\||_0,$ and $\||C \Phi^{-1}\||_0=\||C\||_0$
>
>   _must be a generalized permutation matrix, then such systems are said to be essentially identifiable, up to permutation and scaling._
>
>   Following this definition, if the system is identifiable, the nonsingular matrix $ \Phi$ must be a  generalized permutation matrix to ensure that the number of nonzero components in the system matrices remains unchanged.

---

> ### Author Response · Authors · 2025-11-20
> **Rebuttal #3**
>
> > **While the paper is mathematically rigorous, much of the machinery (Student’s
>  prior for sparsity, hierarchical Bayesian treatment, EM updates, block coordinate descent, even the use of convergence theorems) is largely a synthesis/extension of established sparse Bayesian learning and system identification literature. The main differentiating insight (identifiability under sparsity priors limiting transformations to permutation/scaling) though interesting, is not deeply explored beyond special-case analysis.**
>
> - As discussed in Section 2, while sparse Bayesian learning (SBL) has been applied to learn various dynamical systems, existing studies primarily focus on scenarios where  system states are directly measurable. For linear state-space models (LSSMs), however,  system states  are unobservable, making it infeasible to naturally extend SBL to learn  system matrices. In fact, learning LSSMs with sparse system matrices involves two fundamental problems: **state estimation** and **system identification**.  By exploring the expectation-maximization algorithm, this paper proposes a unified framework that seamlessly integrates the Rauch-Tung-Striebel (RTS) smoother for estimating hidden system states in the expectation step and SBL  for learning sparse system matrices in the maximization step. Consequently, the proposed algorithm enables learning LSSMs with sparse system matrices without prior knowledge of system states.
>
>
> - Based on your comment, we have formalized the definition of identifiability for LSSMs with sparse system matrices. For such systems, the  similarity transformation is restricted  to be a generalized permutation matrix, and  resulting ambiguities are thus limited to permutations and scaling.
>
> >**Although synthetic and 3 real-world datasets are experimented on, comparison to more recent or advanced sparse LSSM learning methods, such as those using Lasso or regularized MLE approaches (e.g., [1]), is missing. Moreover, evaluation is limited mostly to MRE; the case for learning the "correct" topology is largely qualitative except for synthetic scenarios, which could be expanded quantitatively (e.g., with precision/recall on support recovery).**
>
> - We thank the reviewer for pointing to this related work, and  we have carefully reviewed the paper. However, we found that its formulation considers only the state transition equation $ x_t= A x_{t-1}+ B u_{t}+ \varepsilon_{t}$. Because the input $ u_t$ and system state $x_t$ are available, they can directly use Lasso estimator to learn sparse system matrices $ A$ and $ B$.  In contrast, our formulation considers not only the state transition equation $ x_t= A x_{t-1}+ B u_{t}+ \varepsilon_{t}$  but also the observation equation $ y_t=C x_{t}+ D u_{t}+ w_{t}$. For such systems, while the input $ u_t$ and noisy observation  $ y_t$  are given, the system state $ x_t$ remains unobservable, making it challenging to analytically compute the modeling error. Hence, the Lasso approach is not directly applicable  to the LSSMs considered in our work. To address this issue, we propose a learning framework that iteratively  alternates between state estimation via the RTS smoother and system identification via SBL.  As such, we can learn LSSMs with sparse system matrices without prior knowledge of  system states.
>
> - Expect for synthetic scenarios, the underlying system topology is typically unknown, making it infeasible to provide  quantitative metrics such as precision and recall for support recovery. Hence, this paper primarily uses the MRE as a quantitative metric to evaluate the performance of all the algorithms. This choice is also reasonable because we cannot expect a learned LSSM with an incorrect topology to achieve better prediction accuracy in general. Moreover, simulation experiments have demonstrated the superior capability of the proposed algorithm in topology recovery compared to the other algorithms.
>
> **Response to Questions**
>
> >**I would wish author(s) can clearify my concerns around thresholding for sparsity, hyperparameter selection, and comparison to Lasso/regularized MLE-style estimators.**
>
> - See the responses to  Weaknesses.
>
> >**The constraints on similarity transformations in the presence of sparsity-promoting priors are presented for special cases. Can these insights be generalized, or are there pathological systems where the identifiability argument fails?**
>
> - Indeed, there may exist some pathological systems where the identifiability argument does not hold. To address this issue, we have formalized the definition of identifiability for LSSMs with sparse system matrices in Section 3 of the revised version.  Under this definition, if the system is identifiable, the nonsingular matrix $ \Phi$ must be a generalized permutation matrix.

---

> > ### Comment · Reviewer_ttg7 · 2025-11-25
> >
> > Thank author(s) for the detailed response (and updated manuscript). The added computational profiling, runtime comparisons, and discussion of possible acceleration strategies do address my concerns about scalability. The new analyses in Appendix G clarify the thresholding mechanism and robustness. I appreciate the formalization of identifiability for sparse LSSMs, although I still believe that the generality of this condition for more complex or denser systems can benefit from further discussions. Your clarification on why Lasso-style estimators are not directly applicable to hidden-state settings seems reasonable to me too. I feel the rebuttal substantially improves the clarity and completeness of the submission, hence I am happy to raise the evaluation.

---

> > > ### Author Response · Authors · 2025-11-26
> > > **Response to Reviewer ttg7**
> > >
> > > We appreciate the reviewer for raising the score, and your thoughtful feedback has been invaluable in helping us refine the paper. We are sincerely grateful for your recommendation!

---

### Author Response · Authors · 2025-11-30
**Clarification on Our Rebuttal and Scores**

Dear AC and Reviewers,

We sincerely thank you for taking the time to review our paper and for providing valuable evaluations. In light of the unexpected circumstances with OpenReview, we believe it would be helpful to summarize the latest state of the rebuttal prior to the data-leak incident. **Thanks to the prompt feedback from all reviewers, the rebuttal process and score adjustments for our paper had already been completed before the OpenReview bug occurred, as can be verified through the rebuttal history.** During the rebuttal process:

- Reviewer  ttg7 felt the rebuttal substantially improved the clarity and completeness of the submission and  was thus happy to  raise the score from 6 to 8.
- Reviewer jpHJ thought that the rebuttal addressed all of the concerns and therefore updated the score from 6 to 8.
- Reviewer ggGN had no further questions and maintained the initial positive score of 6.

**As a result, the paper's  final scores were  8, 8, and 6 before they were reverted to the initial values.** We respectfully request that you  take this information into consideration when making your final decision. We deeply appreciate your understanding and support.

Best wishes,

The Authors

---

### Author Response · Authors · 2025-12-04
**Author Remarks**

Dear Area Chair and Reviewers,

We sincerely appreciate the Area Chair for the additional time and effort devoted to re-evaluating the rebuttal and  discussion of our manuscript. We are also grateful to all reviewers for the thoughtful suggestions and kind recognition of our contribution.

Below,  we summarize the contributions of the manuscript  and our responses to the reviewers’ main concerns to assist the Area Chair in the assessment.

---

**Contributions:**

- **Algorithmic Innovation:**  Due to the similarity transformation, current algorithms lack the ability to learn  linear state-space models (LSSMs)  with sparse system matrices. To address this issue, we propose an algorithm to learn LSSMs with sparse system matrices by balancing modeling error and model complexity via sparsity-promoting techniques.

- **Theoretical Guarantee:**  Following the Global Convergence Theorem, we  demonstrate that the proposed learning algorithm is guaranteed to converge to a local maximum or saddle point of the objective function composed
of marginal likelihood and prior functions.

- **Experimental Validation:** Experimental results on simulation and real-world datasets demonstrate that the proposed
algorithm outperforms classical ones on learning LSSMs with sparse system matrices. Specifically, only the learned sparse system matrices  of the proposed algorithm preserve the inherent topological structure among variables while yielding the lowest prediction error.

---
**Concerns and Our Responses:**

- **Computational Complexity Analysis :**  We have provided a computational complexity analysis of the proposed algorithm in Appendix F and included a detailed runtime comparison with several standard alternatives across all experiments in the revised version.

- **Threshold Selection:** We have added a discussion on threshold selection  in Appendix G of the revised version, showing that the threshold  can be reliably selected from a wide and stable range.

- **Non-invertible System:** We have added an additional experiment on a non-invertible system in Appendix I.2 of the revised version to further demonstrate the superior performance of the proposed algorithm over classical ones.

- **Structural Definition on Sparse System Matrices:** We have formalized the definition of identifiability for LSSMs with sparse system matrices in Section 3  to ensure that the resulting ambiguities can only be permutations and scaling.

- **Algorithm Robustness:** We have demonstrated the robustness of the proposed algorithm to the threshold in Appendix G. We also conducted two additional sets of experiments to show that the proposed algorithm is robust to the choice of hyperparameters.

---

Finally,  we thank the Area Chair and Reviewers again for the contributions to the community.

Best Regards,

Authors of Paper 1179

---

### Meta-Review · Area_Chair_ubCL · 2026-01-05

**Summary:**

The core contribution of this paper is to develop algorithms for learning sparse system transition matrices (linear dynamical systems) from data. Specifically, the authors identify the challenge where learning latent dynamics suffers from an inherent unidentifiability. To remedy this challenge the authors propose finding the sparsest transition matrix, which they claim also matches the underlying simplicity of many real-world systems.

The authors model the system probabilistically, using Sparse Bayesian Learning with priors that are well established to produce sparse estimates and an expectation-maximization algorithm. The authors provide convergence guarantees, as well empirical results on synthetic data and several real-world datasets.

**Reviewer Concerns:**

The main concerns of the reviewers included
 1) Concerns about the novelty given that the modeling mostly reflects past models in Sparse Bayesian Learning
 2) Concerns about the computational complexity of the algorithms
 3) Concerns about the extent (metrics and algorithms compared to) in the results
 4) Clarifications on some of the concepts that were being introduced/discussed
 5) Applications and potential performance to systems that are non-invertible
 6) Questions about sensitivity to thresholding

The authors responded with a number of improvements including
 1) detailed explanations describing how learning dynamics in the latent space differs from existing work despite the modeling approach being similar
 2) New evaluation of run-time including tests on synthetic data for non-invertible systems and thresholding sweeps to test sensitivity
 3) Additional clarification in the text, including a strict definition of "generalized permutation"

These additions helped alleviate most of the concerns of the reviewers, leaving as outstanding the question of computational complexity. While the complexity concern is noted, and the revision does emphasize this limitation more, it is not solved but is instead a limitation of the method. I don't think this is a terrible issue since all methods have some drawback and this is the price of closed-form solutions.

As an aside, I'd note that the submission might be strengthened by referring to some of the original work on reweighted L1 that also intersects with the SBL style modeling used (see references below). Additionally, there is some relationship with the recent dLDS model which takes a slightly different sparsity-driven approach that also can solve the unidentifiability issue. This model (Mudrik et al. 2024 and Chen et al. 2024) can be considered as dynamics that are sparse in a linear basis, rather than canonically sparse. I don't think that it infringes at all on the work presented, which is very solid, however might help make the background and discussion of how the model fits with the rest of the literature more complete.

Add refs:
 - Candes et al. Enhancing sparsity by reweighted ℓ 1 minimization. Journal of Fourier analysis and applications, 14(5), pp.877-905. 2008
 - Garrigues & Olshausen. Group sparse coding with a laplacian scale mixture prior. Advances in neural information processing systems, 23. 2010
 - O’Shaughnessy, et al. Sparse Bayesian learning with dynamic filtering for inference of time-varying sparse signals. IEEE Transactions on Signal Processing, 68, pp.388-403. 2019
 - Mudrik, N., et al. Decomposed linear dynamical systems (dlds) for learning the latent components of neural dynamics. Journal of Machine Learning Research, 25(59), pp.1-44. 2024.
 - Chen Y., et al. Probabilistic decomposed linear dynamical systems for robust discovery of latent neural dynamics. NeurIPS, 37, pp.104443-104470. 2024.

**Reviewer Scores:**

After the responses from the authors, two of the reviewers had indicated that they would raise their score from a 6 to an 8 (reviewers ttg7 and jpHj), and the third (reviewer ggGN) expressed that they maintained the score of 6. The reviewers all seemed positive both about the paper and the responses from the authors.

---

### Decision · Program_Chairs · 2026-01-26

Accept (Poster)